# Scalable parallel and distributed simulation of an epidemic on a graph

**Guohao Dou** *

School of Computer and Communication Sciences, EPFL, Lausanne, Vaud, Switzerland

* guohao.dou@epfl.ch

**Data Availability Statement:** We release the code and simulation results on Zenodo. Link: https://doi.org/10.5281/zenodo.7750420.

**Funding:** The author(s) received no specific funding for this work.

## Abstract

We propose an algorithm to simulate Markovian SIS epidemics with homogeneous rates and pairwise interactions on a fixed undirected graph, assuming a distributed memory model of parallel programming and limited bandwidth. This setup can represent a broad class of simulation tasks with compartmental models. Existing solutions for such tasks are sequential by nature. We provide an innovative solution that makes trade-offs between statistical faithfulness and parallelism possible. We offer an implementation of the algorithm in the form of pseudocode in the Appendix. Also, we analyze its algorithmic complexity and its induced dynamical system. Finally, we design experiments to show its scalability and faithfulness. In our experiments, we discover that graph structures that admit good partitioning schemes, such as the ones with clear community structures, together with the correct application of a graph partitioning method, can lead to better scalability and faithfulness. We believe this algorithm offers a way of scaling out, allowing researchers to run simulation tasks at a scale that was not accessible before. Furthermore, we believe this algorithm lays a solid foundation for extensions to more advanced epidemic simulations and graph dynamics in other fields.

## 1 Introduction

Large-scale epidemic simulations have become indispensable for high-stakes policymaking in public health. In 2020, the Imperial College report [1] predicted serious consequences in the absence of government actions to control the COVID-19 pandemic, convincing the UK government to issue necessary restrictions. However, large-scale simulations require immense computing power. Just as the authors of [2] put it, building a model on the global scale is a "daunting undertaking". Thanks to the increasing availability of commercial cloud platforms and supercomputers, we can run simulation algorithms on many server nodes. When the task scale is moderate, each computing node runs its instance of the simulation algorithm, and the task is embarrassingly parallel. We only consider the scale at which multiple computing units need to cooperate to finish one instance of the simulation algorithm, meaning that parallelism cannot be attained without communication, and nontrivial algorithm design is needed.

The importance of epidemic simulations and the availability of computing resources calls for algorithmic solutions. We henceforth focus on a highly useful special case to build a solid foundation for future extensions, and we define the scope of the problem as follows.

**Competing interests:** The authors have declared that no competing interests exist.

- By "Markovian SIS epidemics", we consider epidemics with an SIS (susceptible-infected-susceptible) compartmental model where all times to infection and times to recovery follow exponential distributions. In an SIS model, each individual, represented as a vertex in the graph, can either be in an *infected* (INF) state or a *susceptible* (SUS) state. Susceptible vertices neighboring infected vertices may transition into the infected state through an *infection* event. Infected vertices may transition into the susceptible state through a *recovery* event. Non-Markovian epidemic models exist and have become a rich field of study. See [3, 4].

- By "homogeneous rates", we mean that all exponentially distributed times to infection/recovery have the same infection/recovery rates, as opposed to heterogeneous rates, studied in [5, 6].

- By "pairwise interactions", we mean that all infection events through edges happen independently, in contrast to the cooperative infection model in [4]. Dynamics beyond pairwise interactions have also become an emerging field [7].

- By "a fixed graph", we mean that the graph structure on which the epidemic dynamics occur is fixed throughout the simulation. A body of literature exists for the scenario where the graph structure is allowed to vary, sometimes adaptively. See [8–11].

- By "distributed memory model of parallel programming", we mean computing units can communicate by sending messages alone. It differs from the shared memory model of parallelism. We ensure that the algorithm lends itself well to extreme-scale simulation tasks where researchers have to scale out in a loosely connected cluster of machines.

- By "limited bandwidth", we put an upper bound on the bandwidth usage, which tends to become the performance bottleneck.

We pick the simplest epidemic model (SIS) as an example of our algorithm. It will become clear that our algorithm can be easily generalized to other compartmental models. These limitations are as restrictive as they are instructive for future extensions. For now, they simplify the presentation and analysis of the algorithm.

Efforts have been made to simulate spreading dynamics on graphs. By now, there are mainly two schools of thought: *time-driven simulations* (TDS) [1, 2, 8, 12] and *event-driven simulations* (EDS) [13–16].

TDS proceeds in epochs of time steps of fixed size $\Delta$. Let the recovery rate be $\gamma$ and the infection rate $\beta$. In an epoch of a TDS algorithm, the algorithm scans through all vertices of the graph. If a vertex $v$ is infected, it recovers with probability $1 - \exp(-\gamma\Delta)$ at the end of the epoch. If a vertex $v$ with $n_v$ infected neighbors is susceptible at the beginning of the epoch, then it gets infected with probability $1 - \exp(-n_v\beta\Delta)$ at the end of the epoch. TDS is easy to implement and parallelize and can be fairly efficient if a large $\Delta$ is chosen, which explains why it is popular among researchers and practitioners [1, 2, 8, 12]. However, arguments can also be made against TDS. Firstly, the system's state can only be observed at multiples of $\Delta$. Secondly, as pointed out in [17], discretization introduces a significant bias from continuous formulation such as the master equation, a bias worsened by large $\Delta$.

EDS proceeds by hopping from one event to another, usually by maintaining a queue of events ordered by their scheduled times of occurrence and processing these events in this order. Unlike TDS, EDS allows events to occur at any time, not just multiples of $\Delta$. The occurrence of an event may schedule new events or cancel existing events, with the time between the scheduling and occurrence of an event, which we call the "pending time", following a certain distribution (exponential distribution in our case). When the infection of a vertex $v$ takes place at time $t$, a recovery event of vertex $v$ gets scheduled at time $t + T_{\text{rec}}$ where $T_{\text{rec}}$ follows an

exponential distribution with rate $\gamma$. When the recovery of a vertex $v$ with $n_v$ infected neighbors takes place at time $t$, an infection event of vertex $v$ gets scheduled at time $t + T_{inf}$ where $T_{inf}$ follows an exponential distribution with rate $n_v\beta$. EDS is faithful to the continuous formulation, and the system's state can be observed at any time. However, EDS is sequential by design since the events must be processed one by one. Tremendous efforts have gone into parallelizing EDS, giving rise to the PDES (Parallel Discrete Event Simulation) community [18]. Two main methods exist in the PDES community, proposing two different ways of preventing events from occurring out of order. The first, often referred to as the CMB algorithm (Chandy, Misra, and Bryant) [19], relies on the crucial assumption that pending times have a positive lower bound and the algorithm blocks execution if out-of-order event processing is possible. The second, proposed in [20], allows out-of-order errors but employs a rollback mechanism to undo the errors. People have also adopted PDES techniques in epidemic simulations before [21], even though the authors of [21] treat each location as a logical process, not each individual, making their algorithm less generic than simulating epidemics on graphs. We believe that methods from the PDES community are not well-suited for simulating dynamical processes on graphs. On the one hand, pending times do not have a lower bound, making the CMB algorithm inapplicable and the algorithm from [20] arbitrarily inefficient. On the other hand, each vertex needs to be its own logical process, which brings about significant overhead from various sources, such as communication and operating system scheduling.

We propose in this article an algorithm that strikes a balance between TDS and EDS and enables the trade-off between parallelism and faithfulness (faithfulness to the continuous formulation of the dynamical system). The two can be viewed as two extremes of our algorithm. In fact, when the number of partitions is one ($M = 1$ as we will see later), our algorithm is reduced to classic sequential EDS, which we use as a baseline in our experiments. Our algorithm introduces parallelism through two techniques: partitioning and summary. We partition the graph and allow each part to run its own copy of EDS. However, partitioning without communication causes each part to observe a stale version of their neighbors' states, leading to highly biased simulation results. To solve this problem, we take inspiration from TDS and force all parts to synchronize periodically, exchanging only a summary of its local information, which saves precious bandwidth.

We study our algorithm's complexity, scalability, and faithfulness through theoretical and experimental means. We discover that, under mild assumptions, our algorithm scales well with the number of computing units and preserves distinctive features of the dynamical system despite a highly adversarial parameter setup.

The paper is organized as follows. First, we introduce basic notations, definitions, and assumptions in Section 2. Second, we describe the algorithm in Section 3 and offer an implementation of our algorithm in the form of pseudocode in Section 7. Third, we study the algorithm theoretically in Section 4 and experimentally in Section 5. Finally, Section 6 sums up the article and discusses the limitations and extensions of the algorithm.

## 1.1 Related work

Modeling epidemic processes is an important subject throughout human history. As early as 1766, Daniel Bernoulli publishes one of the earliest compartmental models about smallpox [22]. In 1927, Kermack et al.[23] publish what is commonly known as the Kermack-McKendrick theory, a seminal contribution that models epidemic dynamics in a homogeneous population with a system of differential equations. Their work gives rise to a rich body of literature on population dynamics that discusses complex compartmental transitions [24] and applications on real-world epidemic data [25].

In recent years, epidemic modeling on graph structures has attracted intellectual attention. Instead of dynamics in a homogeneous well-mixed population, which can usually be described by a handful of differential equations, researchers in this field focus on compartmental models on graphs, where each vertex in the graph is in one of the predefined compartments/states, and state transitions take place as a result of interactions among neighboring vertices. In this setting, an exact description of the dynamics requires an enormous number of differential equations, a number that typically scales exponentially with the number of vertices in the graph. Understanding the mathematical intractability, researchers explore various methods of simplification such as the mean-field approximation, resulting in a wealth of literature. See [26] for a comprehensive overview.

Because of the mathematical difficulty, many researchers and practitioners resort to numerical simulations. We have discussed copiously the two main streams of methods: EDS and TDS. There are also ad-hoc or agent-based methods of epidemic simulation, such as [1, 21, 27–29]. Many of these ad-hoc methods do employ parallel computation. However, their model adequacy comes at the cost of model explainability and generality.

Our algorithm can be viewed as a part of the broader effort in graph data processing. Many existing systems for parallel graph processing can be repurposed to implement our algorithm, such as [30, 31]. Even though we simply use METIS [32] for graph partitioning, a community detection algorithm [33, 34] can also be a reasonable choice. Moreover, because the ultimate goal of epidemic simulations is to predict the future, it is plausible that we can feed simulation trajectories to a graph neural network [35, 36] as training data and use the graph neural network for fast predictions later on.

## 2 Problem setup

We have as input

- a fixed undirected graph $G = (V, E)$ with initial vertex states (INF or SUS); INF for infected and SUS for susceptible;

- a partition of $V = V_1 \cup V_2 \cup V_3 \cup \cdots \cup V_M, \forall i \neq j, V_i \cap V_j = \emptyset, \forall i, V_i \neq \emptyset$;

- an SIS epidemic model with recovery rate $\gamma$ and per-edge infection rate $\beta$;

- epoch length $\Delta$; simulation task horizon $H$.

The epoch length $\Delta$ plays a similar role as the fixed time step in TDS. The simulation task horizon $H$ specifies how much time in the simulated world should have elapsed by the time the simulation algorithm terminates.

We make the following assumptions on the input:

- $M^2$ is $\mathcal{O}(|V|)$. If we treat $M$ as a function of $|V|$, $M(|V|)$ satisfies

$$\lim_{|V| \to \infty} \frac{[M(|V|)]^2}{|V|} = 0.$$

- $H, \Delta, \beta, \gamma$ are constant functions of $|V|$ and $M$. $H, \Delta, \beta, \gamma \in \mathbb{R}^+$.

- The mean degree $\langle D \rangle$ of $G$ exists and is a constant function of $|V|$ and $M$.

We simulate the spread of Markovian SIS epidemics with homogeneous rates and pairwise interactions. A simulation algorithm produces a trajectory of the form

$$(t_0, e_0), (t_1, e_1), \ldots, (t_L, e_L), \quad 0 \leq t_0 < t_1 < t_2 < \cdots < t_L \leq H,$$

where $e_0, e_1, \ldots e_L$ are events that take place at $t_0, t_1, \ldots t_L$ in the simulated world. We refer to the production of one such trajectory as one simulation instance.

For this specific application, events are either infections or recoveries. A recovery event Rec($v$) is scheduled when vertex $v$ changes its state from susceptible (SUS) to infected (INF). When Rec($v$) occurs, the state of $v$ becomes susceptible. An infection event Inf($u \rightarrow v$) is scheduled when the state change ($u$ susceptible XOR $v$ infected) $\rightarrow$ ($u$ infected AND $v$ susceptible) occurs. When Inf($u \rightarrow v$) occurs, the state of $v$ becomes infected. We keep track of the origin of infection because we want to differentiate between these infecting events, and only the event with the smallest scheduled time may occur, upon which all others will be canceled.

Suppose we have $M$ processes $P_1, \ldots, P_M$ interconnected by a network, each with sufficient and fast local memory and a dedicated computing unit. We map each $V_i$ to $P_i$, where we keep in memory

- $\forall v_i \in V_i$, the state of $v_i$ and the list of its neighbors;

- data structures used for computation local to $P_i$.

If a vertex $v$, its state, and the list of its neighbors reside in the memory of process $P$, we say $v$ is a vertex on process $P$ (or, local to process $P$). For the rest of this article, we abuse the notation and use the symbol of process $P$ to denote the set of vertices on it in the context of set operations.

Note that the processes $P_1, \ldots, P_M$ also form a directed graph $(\mathcal{V}, \mathcal{E})$, where

$$\mathcal{V} = \{P_1, \ldots, P_M\}, \quad \mathcal{E} = \{(P_i, P_j) | i \neq j, \exists u \in P_i, v \in P_j, (u, v) \in E\}.$$

We denote the neighbors of vertex $v$ as $\mathcal{N}(v) = \{u | (u, v) \in E\}$. We denote the neighbors of process $P$ as $\mathcal{N}(P) = \{Q | (P, Q) \in \mathcal{E}\}$.

Each of $P_1, \ldots, P_M$ can only efficiently access its memory. Therefore, they have to communicate via message passing. A channel exists for every $(P_i, P_j) \in \mathcal{E}$, where $P_i$ is the sender of messages and $P_j$ the receiver (note that $(P_i, P_j)$ and $(P_j, P_i)$ are two different channels). Suppose that each channel can hold at most one 64-bit integer at a time, and every integer message communicated in the algorithm can be expressed as a 64-bit integer. That is, the system is supposed to crash if $P_i$ puts an integer in the $(P_i, P_j)$ channel before $P_j$ takes the previous integer message out. We refer to a matched pair of sending and receiving as one message. We measure communication costs by the number of messages sent from all processes throughout one simulation instance.

We need an algorithm that can produce the aforementioned trajectory while sending $\mathcal{O}(|V|)$ messages.

## 3 Algorithm

We present and discuss the algorithm in this section. We begin by describing the algorithm on a high level. We provide the pseudocode of the algorithm in Section 7. We also compare it to other existing implementations for parallel processing.

Each process deserializes its assigned part of the adjacency list and initializes its data structures, including but not limited to the event queue essential to event-driven simulations. After initialization, the algorithm proceeds in iterations, which we refer to as "epochs". Given simulation horizon $H$ and epoch length $\Delta$, the loop runs for $\lceil H/\Delta \rceil$ times, after which the algorithm terminates.

In each epoch, a process computes, based on its local memory, what will be needed by other processes for their local computation, after which it sends these messages asynchronously. It

then waits for the messages from other processes to begin its own local computation. After finishing its local computation, it waits on the acknowledgements from other processes that the messages it has sent earlier has been received and then the cycle repeats.

Two key questions are manifested. Firstly, what information needs to be sent to other processes. Secondly, what local computation needs to take place.

The first question determines bandwidth usage. We choose to send merely a summary statistic from process $P$ to process $Q$, which acts as a proxy of $P$ on $Q$. If a vertex $v$ on process $P$ has at least one neighbor on process $Q$, we say $v$ is *bordering* $Q$. Our choice of the summary statistic for this article is the number of infected vertices on $P$ bordering $Q$, denoted as $n_{P \to Q}$. Let the number of vertices on $P$ bordering $Q$ be $N_{P \to Q}$. Because we assume a fixed network, $N_{P \to Q}$ only needs to be computed once during initialization. We then compute the probability of finding an infected vertex on the $(P \to Q)$ border as $n_{P \to Q}/N_{P \to Q}$.

Let $n_{P \to Q}^l$, $l \in \mathbb{N}^+$ be the $l$-th integer that process $P$ sents to $Q$. We have the IO automaton in Fig 1, which illustrates the messaging pattern and the iterative nature of the algorithm.

The second question is partially answered by classic EDS. On top of the event-driven scheme that we have reviewed in the Introduction, we incorporate the information from $n_{Q \to P}$ by introducing a new type of event, $\text{Inf}(Q \to v)$, for susceptible vertex $v \in P$ bordering $Q$. Let $n_{Q \to v}$ be the number of $v$'s neighbors on $Q$. The rate of event $\text{Inf}(Q \to v)$ is computed as

$$
\begin{aligned}
n_{Q \to v} \cdot \frac{n_{Q \to P}}{N_{Q \to P}} \cdot \beta \quad &= \text{number of neighbors of } v \text{ on process } Q \\
&\times \text{probability of these neighbors being infected} \\
&\times \beta.
\end{aligned}
$$

We effectively conduct a mean-field approximation on the border. We summarize this alternating pattern of communication and computation in Fig 2.

Notably, upon receiving a new value for $n_{Q \to P}$, any event on $P$ of the form $\text{Inf}(Q \to v)$ needs to be updated with the new rate by generating a new scheduled time of occurrence.

We offer our implementation in the form of pseudocode in Section 7.

This pattern of communication and computation bears a great resemblance to the "ghost cell" method commonly employed in numerical method and high-performance computing communities [37–41]. The key difference is that instead of sending the whole border area (shaded rectangles in Fig 2) over the network, we exploit the nature of event-driven simulations and send only a summary statistic of the border, vastly reducing bandwidth usage. In addition to the bandwidth reduction, in practice, small messages also allow MPI to use the eager protocol, which enables better overlap of communication and computation [42], compared to the expensive rendezvous protocol designed for large messages.

Besides our implementation in Section 7, many other systems for parallel processing can be used to implement our algorithm. One of the earliest attempt to model this pattern is Bulk Synchronous Parallel (BSP) [43], which consists of many supersteps. Within each superstep, processors engage in local computation, communicate with each other, and wait on a global synchronizing barrier, after which the next superstep may begin. These supersteps play the same role as our epochs, except that our implementation does not rely on a global barrier, which is expensive in practice. Instead, synchronization in our implementation happens through data dependency stipulated by the process graph $(\mathcal{V}, \mathcal{E})$. BSP also uses remote memory access while our implementation uses message passing, which is more commonly supported in terms of hardware.

Another system inspired by BSP is Pregel [30], which is a vertex-centric framework to conduct large-scale graph processing. A processor in BSP is mapped to a vertex in Pregel, which

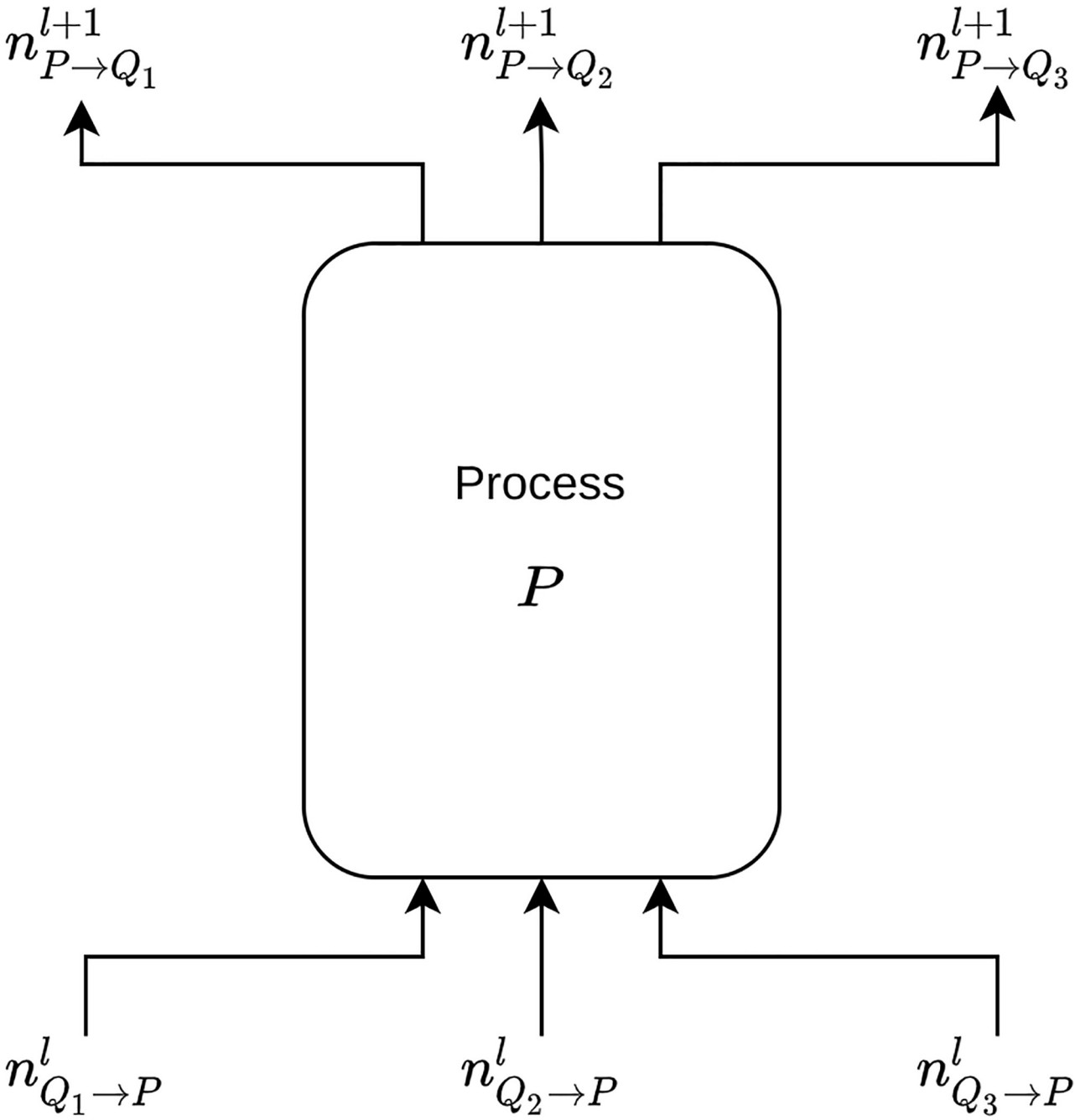

**Fig 1. The IO automaton of process $P$.** We assume that process $P$ has neighboring processes $Q_1$, $Q_2$, $Q_3$.

sends/receives messages to/from other vertices and modifies its state accordingly. Like BSP, Pregel also has a global barrier, implemented as the master in the master-worker architecture, which additionally takes care of fault tolerance. Our algorithm can be readily implemented under the Pregel framework, with the graph being the process graph $(\mathcal{V}, \mathcal{E})$.

[31] offers a good review of Pregel-like parallel processing systems. All examined in [31] but GraphLab [44] adopt the BSP model, with global barriers for synchronization. GraphLab

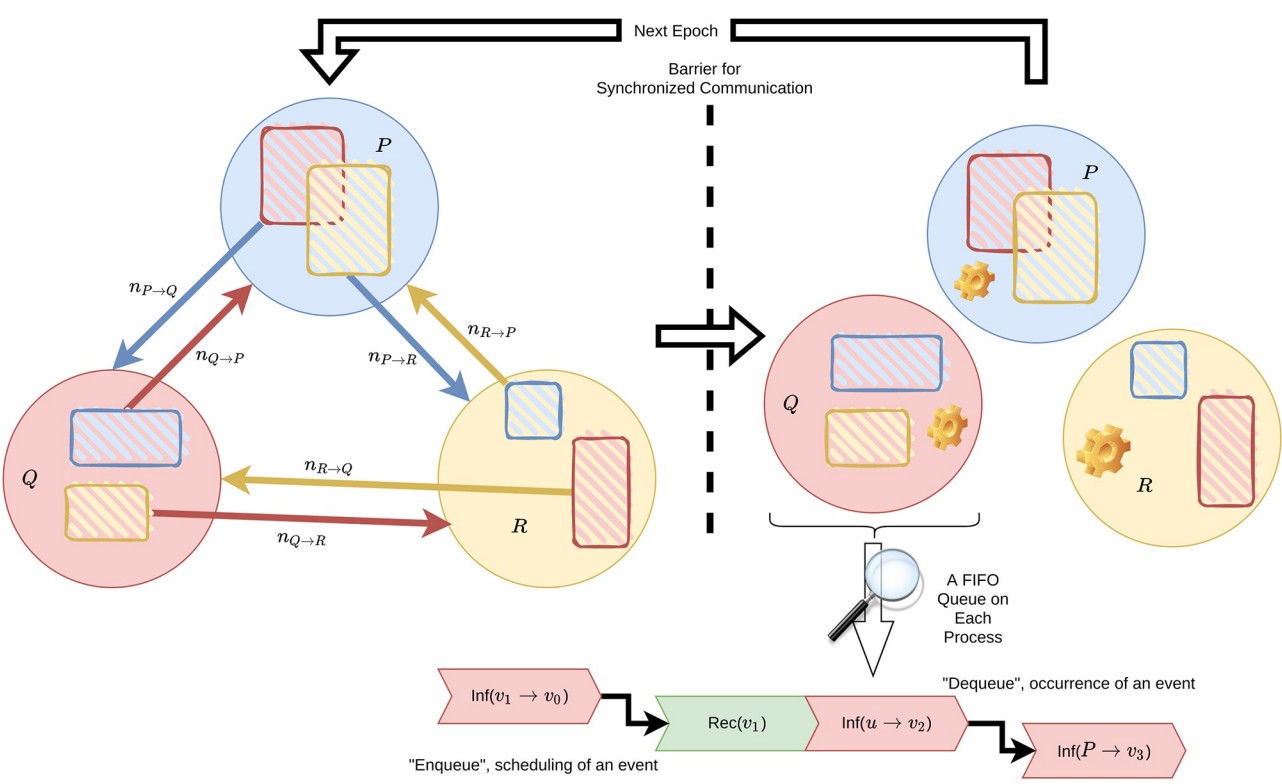

**Fig 2. An example with 3 interconnected processes.** Each circle stands for a process. The blue rectangle in the red circle stands for the subset of vertices on $Q$ that border $P$. $n_{Q \to P}$ is computed based on this subset and sent to $P$ for $P$'s local computation of the next epoch. Each process maintains its own priority queue, which sorts events by their scheduled times of occurrence and keeps picking the most imminent event.

[44] has an asynchronous mode with no global barrier. However, this asynchronous mode only ensures serializability, which precludes race conditions but does not gurantee the iterative semantics of our implementation and other BSP-based systems. Moreover, GraphLab uses shared memory for locking to ensure serializability, which hampers performance due to intensive lock contention according to [31].

# 4 Theoretical results

## 4.1 Complexity

We discuss in this section the complexity of algorithm 5 in S1 File. We make some simplifying assumptions on the independence of random variables to make the problem mathematically tractable.

- The state of a vertex $v$ can be modeled by a stochastic process, $S_v(t)$, where

$$\Pr\{S_v(t) = 1\} = \Pr\{v.\text{state@}t = \text{INF}\},$$
$$\Pr\{S_v(t) = 0\} = \Pr\{v.\text{state@}t = \text{SUS}\}.$$

We assume that $\forall t \in [0, H], \forall v \in V, \mathbb{E}[S_v(t)] = P_I(t)$. We also assume that $S_v(t)$ is independent of its neighborhood. More precisely, let $v$ be a vertex with $d$ neighbors, $v_1, v_2, \ldots, v_d$.

We assume that the factorization

$$\Pr\{S_v = s, S_{v_1}(t) = s_1, \ldots, S_{v_d}(t) = s_d\}$$
$$= \Pr\{S_v = s\} \cdot \Pr\{S_{v_1}(t) = s_1, \ldots, S_{v_d}(t) = s_d\}$$

is legitimate. Let $F : \{0, 1\}^d \to \mathbb{R}$ be a function defined on its neighborhood. This assumption implies that the factorization

$$\mathbb{E}[S_v(t) \cdot F(S_{v_1}(t), \ldots, S_{v_d}(t))] = \mathbb{E}[S_v(t)] \cdot \mathbb{E}[F(S_{v_1}(t), \ldots, S_{v_d}(t))]$$

must also be legitimate.

- The degree of a vertex $v$, $D_v$, can be modeled by a random variable taking values from $\mathbb{N}$. We assume that $\{D_v\}_{v \in V}$ are i.i.d. random variables with $p_d = \Pr\{D_v = d\}$ and $\langle D \rangle = \sum_{d=0}^{\infty} p_d d$.

- The process membership of a vertex $v$, $\Pi_v$, can be modeled by a random variable taking values from $\{1, 2, \ldots, M\}$. We assume that $\{\Pi_v\}_{v \in V}$ are i.i.d. random variables with

$$\forall v \in V, \forall j \in \{1, 2, \ldots, M\}, \Pr\{\Pi_v = j\} = 1/M.$$

- $\forall t \in [0, H], \forall v \in V$, the collection of random variables $\{S_v(t)\} \cup \{D_v\}_{v \in V} \cup \{\Pi_v\}_{v \in V}$ consists of independent random variables.

We now review and justify these assumptions. We present the scenarios where these assumptions hold and point out directions where they may be loosened.

- The assumption on vertex state independence echoes the mean-field assumption in [26, 45, 46]. It is common in the literature to assume independence among all vertex states, which is too strong an assumption for what our analysis requires. Also, because infected vertices are more likely to be neighboring each other in actual epidemic processes, assuming vertex state independence leads to an overestimated count of (INF, SUS) edges, which suits our purpose of showing an upper bound. This assumption works well for dynamic contact graphs where the neighborhood keeps changing and complete graphs where the neighborhood is the entire graph. In both cases, the neighborhood is a good representative of all vertices. In [26, 47], analyses of refined models beyond the one-vertex mean-field approximation are provided.

- The assumption on vertex degree independence comes from the configuration model (without degree correlation) where a random number of "stubs" are generated on each vertex and then randomly connected [48]. The configuration model is a powerful graph generation method because it allows one to specify arbitrary degree distribution. It works well when degree correlation is shown to be negligible. We are aware that degree correlation does exist in real-world graph structures [49], and people have studied epidemic dynamics on graphs with degree correlation [50].

- This assumption on process membership is a consequence of partitioning the graph uniformly at random, which is a sensible partitioning scheme without prior knowledge about the graph structure. In practice, one may choose a graph partitioning software that reduces the number of interprocess edges, such as [51].

- The assumption that $\{D_v\}_{v \in V} \cup \{\Pi_v\}_{v \in V}$ are independent makes sense as long as graph partitioning is done without considering graph topology. The assumption that $S_v(t)$ and $\{\Pi_v\}_{v \in V}$ are independent may not hold for highly irregular graphs, where some processes are significantly more infected than others. However, because we have assumed uniform graph

partitioning, the chances of one part being highly different from the others are slim. The assumption that $S_v(t)$ and $\{D_v\}_{v \in V}$ are independent works well when the degree distribution is highly centralized around its mean and is trivially true if we are considering a regular graph. As pointed out in [52], the assumption that $\{S_v(t), D_v\}$ are independent is unrealistic because intuitively, vertices with more neighbors ought to have a higher chance of catching the disease.

Any deviation of the input from these assumptions will inevitably hurt the validity of our analysis. Nevertheless, we decide to stick to this classical set of assumptions to attain an early result on complexity analysis and lay a foundation for future refinement.

Take an arbitrary process $P$. We denote the stochastic process $N(t)$ as the number of events in the queue of process $P$ at time $t$ in the simulated world. We denote the total rate $\Lambda(t)$, also a stochastic process, as the sum of rates of all infection events and recovery events in the queue of process $P$ at time $t$ in the simulated world.

Let $\mu_v$ be the number of distinct processes that vertex $v$ on $P$ is bordering. If we pick uniformly at random $\pi_1, \pi_2, \ldots, \pi_{D_v}$ from the alphabet $\{1, 2, \ldots, M\}$ with replacement, then $\mu_v = |\{\pi_1, \pi_2, \ldots, \pi_{D_v}\} \setminus P|$. By construction, $\mu_v \leq \min\{D_v, M - 1\}$. Finding the distribution of $\mu_v$ is nontrivial, but its expectation can be easily shown as

$$\mathbb{E}[\mu_v] = (M - 1)\left[1 - \psi\left(1 - \frac{1}{M}\right)\right],$$

where $\psi(x) = \sum_{d=0}^{\infty} p_d x^d$ is the generating function of the degree distribution.

Queue operations dominate the computational costs. Therefore, we state the number of queue operations in proposition 1.

**Proposition 1**. *The number of operations (up to constant factors) incurred by the queue data structure on process $P$ in one simulation instance, denoted as random variable $\tilde{C}$, can be expressed as*

$$\tilde{C}_1 = \int_0^H dt\, \Lambda(t) D_v \log N(t),$$

$$\tilde{C}_2 = \sum_{k=0}^{H/\Delta - 1} \log N(k\Delta) \sum_{v \in V} \mathbb{1}\{\Pi_v = P\} \cdot \mathbb{1}\{S_v(k\Delta) = 0\} \cdot \mu_v,$$

$$\tilde{C} = \tilde{C}_1 + \tilde{C}_2.$$

*where we assume without loss of generality that $H/\Delta$ yields an integer.*

$\tilde{C}_1$ is the number of queue operations within epochs. In each infinitesimal $dt$, an event occurs with probability $dt\Lambda(t)$, upon which it flips the state of vertex $v$ and examines its neighbors, causing at most $D_v$ queue push or remove operations, each with complexity $\log N(t)$. $\tilde{C}_2$ is the number of queue operations between epochs. Every susceptible vertex $v$ on $P$ bordering some other process $Q$ causes a queue fix operation of event $\text{Inf}(Q \to v)$ upon receiving an updated border infection count $n_{Q \to P}$. A queue fix at the beginning of the $(k + 1)$-th epoch has complexity $\log N(k\Delta)$. Each susceptible vertex $v$ causes $\mu_v$ queue fixes.

Where proposition 1 gains precision, it loses mathematical tractability. The key issue lies in the fact that $\Lambda(t)$, $D_v$, and $\log N(t)$ are not independent random variables, nor are $\log N(k\Delta)$, $\Pi_v$, $S_v(k\Delta)$, and $\mu_v$. We resort to more approximations to counter this issue and attain a good qualitative description of the scaling behavior. We define $\mathcal{Q}(|V|, M)$ as a function of $|V|$ and $M$ that satisfies

$$\forall t, \lim_{M \to \infty} \Pr\{\log N(t) \geq \mathcal{Q}(|V|, M)\} = 0,$$

meaning that $\mathcal{Q}$ is an upper bound of queue operation complexity almost surely when $M \to \infty$. We also assume that $D_v$ is concentrated around its mean, allowing us to use $\langle D \rangle$ in its stead. For $\mathbb{1}\{S_v(k\Delta) = 0\}$, we simply upper-bound it by 1.

After these, we obtain an expression for the approximate complexity $\mathcal{C}$.

**Definition 1**.

$$\mathcal{C}_1 = \mathcal{Q}\langle D \rangle \int_0^H dt \Lambda(t),$$

$$\mathcal{C}_2 = \mathcal{Q} \sum_{k=0}^{H/\Delta-1} \sum_{v \in V} \mathbb{1}\{\Pi_v = P\} \mu_v,$$

$$\mathcal{C} = \mathcal{C}_1 + \mathcal{C}_2.$$

We will focus on providing an upper bound for $\mathbb{E}\mathcal{C}$.

**Theorem 1**.

$$\forall t, \mathbb{E}[N(t)] \leq N^* = \frac{|V|}{4\langle D \rangle M^2} \left( \langle D \rangle + M^2 \right)^2. \tag{1}$$

$$\forall t, \mathbb{E}[\Lambda(t)] \leq \Lambda^* = \frac{|V|}{4\beta\langle D \rangle M} \left( \beta\langle D \rangle + \gamma \right)^2. \tag{2}$$

*Proof.* Let $v$ be an arbitrary vertex on process $P$. The number of infected neighbors on $P$ of vertex $v$, $\eta_v(t)$, can be expressed as

$$\eta_v(t) = \sum_{d=1}^{D_v} \mathbb{1}\{\Pi_{v_d} = P\} \cdot S_{v_d}(t),$$

where $v_d$ is the $d$-th neighbor of vertex $v$.

$$\begin{aligned}
\mathbb{E}[\eta_v(t)] &= \sum_{d=1}^{\infty} p_d \sum_{k=1}^{d} \mathbb{E}[\mathbb{1}\{\Pi_{v_k} = P\} \cdot S_{v_k}(t)] \\
&= \sum_{d=1}^{\infty} p_d \sum_{k=1}^{d} \mathbb{E}[\mathbb{1}\{\Pi_{v_k} = P\}] \cdot \mathbb{E}[S_{v_k}(t)] \\
&= \sum_{d=1}^{\infty} p_d d \frac{1}{M} P_I(t) = \frac{P_I(t)}{M} \langle D \rangle.
\end{aligned}$$

The number of events of the form $\text{Inf}(u \to v)$, $u, v \in P$, denoted as stochastic process $N_{inf,local}(t)$, is

$$\begin{aligned}
N_{inf,local}(t) &= \sum_{v \in V} \mathbb{1}\{\Pi_v = P\} \cdot \mathbb{1}\{S_v(t) = 0\} \cdot \eta_v(t), \\
\mathbb{E}[N_{inf,local}(t)] &= \sum_{v \in V} \mathbb{E}[\mathbb{1}\{\Pi_v = P\} \cdot \mathbb{1}\{S_v(t) = 0\} \cdot \eta_v(t)] \\
&= \sum_{v \in V} \mathbb{E}[\mathbb{1}\{\Pi_v = P\}] \cdot \mathbb{E}[\mathbb{1}\{S_v(t) = 0\}] \cdot \mathbb{E}[\eta_v(t)] \\
&= |V| \cdot \frac{1}{M} \cdot (1 - P_I(t)) \cdot \frac{P_I(t)}{M} \langle D \rangle \\
&= \frac{|V|\langle D \rangle}{M^2} P_I(t)(1 - P_I(t)),
\end{aligned}$$

which makes

$$\mathbb{E}[\Lambda_{inf,local}(t)] = \beta \frac{|V|\langle D\rangle}{M^2} P_I(t)(1 - P_I(t)).$$

The number of events of the form $\text{Inf}(Q \to v)$, $Q \neq P$, $v \in P$, denoted as stochastic process $N_{inf,remote}(t)$, is

$$N_{inf,remote}(t) = \mathbb{1}\{P_I(t) \neq 0\} \sum_{v \in V} \mathbb{1}\{\Pi_v = P\}\mathbb{1}\{S_v(t) = 0\}\mu_v, \tag{3}$$

$$\mathbb{E}[N_{inf,remote}(t)] = \mathbb{1}\{P_I(t) \neq 0\} \frac{|V|}{M} (1 - P_I(t))\mathbb{E}[\mu_v] \tag{4}$$

$$\leq \mathbb{1}\{P_I(t) \neq 0\} \frac{|V|(M-1)}{M} (1 - P_I(t)). \tag{5}$$

The total rate from other processes, $\Lambda_{inf,remote}(t)$, satisfies

$$\Lambda_{inf,remote}(t) \quad = \sum_{v \in V} 1\{\Pi_v = P\} \cdot 1\{S_v(t) = 0\} \sum_{d=0}^{D_v} 1\{\Pi_{v_d} \neq P\} P_I(t)\beta,$$

$$\mathbb{E}[\Lambda_{inf,remote}(t)] \quad = \frac{|V|(M-1)}{M^2} \beta\langle D\rangle P_I(t)(1 - P_I(t)).$$

The number of events of the form $\text{Rec}(v)$, $v \in P$, denoted as the stochastic process $N_{rec}(t)$, is

$$N_{rec}(t) = \sum_{v \in V} 1\{\Pi_v = P\}1\{S_v(t) = 1\}, \qquad \mathbb{E}[N_{rec}(t)] = \frac{|V|}{M} P_I(t),$$

which makes

$$E[\Lambda_{rec}] = \gamma \frac{|V|}{M} P_I(t).$$

Then we have

$$\mathbb{E}[N(t)] \quad = \mathbb{E}[N_{inf,local}(t) + N_{inf,remote}(t) + N_{rec}(t)]$$

$$\leq -\frac{|V|\langle D\rangle}{M^2} P_I(t)^2 + \frac{|V|(\langle D\rangle + M(2-M))}{M^2} P_I(t) + \frac{|V|(M-1)}{M},$$

which takes maximum when $P_I = \frac{1}{2} - \frac{1}{2}\frac{M(M-2)}{\langle D\rangle}$, and the maximum is

$$\frac{|V|\langle D\rangle}{4M^2} + \frac{|V|}{2} + \frac{|V|(M-2)^2}{4\langle D\rangle} \leq \frac{|V|(\langle D\rangle + M^2)^2}{4\langle D\rangle M^2} = N^*.$$

And we have

$$\mathbb{E}[\Lambda(t)] \quad = \mathbb{E}[\Lambda_{inf,local}(t) + \Lambda_{inf,remote}(t) + \Lambda_{rec}(t)]$$

$$= -\frac{|V|}{M} \beta\langle D\rangle P_I(t)^2 + \frac{|V|}{M} (\beta\langle D\rangle + \gamma)P_I(t),$$

which takes maximum when $P_I = \frac{1}{2} + \frac{1}{2}\frac{\gamma}{\beta\langle D\rangle}$, and the maximum $\Lambda^*$ is

$$\Lambda^* = \frac{|V|}{M}\left(\frac{\beta\langle D\rangle}{4} + \frac{\gamma}{2} + \frac{\gamma^2}{4\beta\langle D\rangle}\right) = \frac{|V|}{4\beta\langle D\rangle M}(\beta\langle D\rangle + \gamma)^2.$$

Note that theorem 1 does not rely on the assumption that $M^2$ is $\mathcal{O}(|V|)$.

**Lemma 1.** $\log N(t) < 4\log\left(\frac{|V|}{M}\right)$ *almost surely as* $M \to \infty$.

*Proof.* By assuming that $M^2$ is $\mathcal{O}(|V|)$, we have

$$\lim_{M\to\infty}\frac{M^2}{|V|} = 0,$$

where we treat $|V|$ as the inverse function of $M(|V|)$.

Using Markov inequality, we have $\forall a$,

$$\Pr\{\log N(t) \geq \log a\} = \Pr\{N(t) \geq a\} \leq \frac{\mathbb{E}[N(t)]}{a}.$$

With a change of variable $b = \log a$,

$$\Pr\{\log N(t) \geq b\} \leq \frac{\mathbb{E}[N(t)]}{e^b} \leq \frac{N^*}{e^b}.$$

Take $b = 4\log(|V|/M)$, and we obtain

$$\begin{aligned}\Pr\{\log N(t) \geq 4\log(|V|/M)\} &\leq \frac{N^*}{(|V|/M)^4}\\ &= \frac{\langle D\rangle M^2}{|V|^3} + \frac{M^4}{2|V|^3} + \frac{M^6}{4\langle D\rangle|V|^3},\end{aligned}$$

which implies that

$$\lim_{M\to\infty}\Pr\{\log N(t) \geq 4\log(|V|/M)\} = 0.$$

Lemma 1 offers us a candidate for $\mathcal{Q}$, namely,

$$\mathcal{Q}(|V|, M) = 4\log\left(\frac{|V|}{M}\right).$$

**Theorem 2.**

$$\lim_{M\to\infty}\mathbb{E}\mathcal{C} \leq 4H\frac{|V|}{M}\log\left(\frac{|V|}{M}\right)\left[\frac{(\beta\langle D\rangle + \gamma)^2}{4\beta} + \frac{\langle D\rangle}{\Delta}\right].$$

*Proof.* First we have

$$\mathbb{E}\mathcal{C}_1 \leq \mathcal{Q}\langle D\rangle H\max_{0\leq t\leq H}\mathbb{E}[\Lambda(t)] \leq \mathcal{Q}H\langle D\rangle\Lambda^*.$$

Then we have

$$\mathbb{E}\mathcal{C}_2 = \mathcal{Q}\sum_{k=0}^{H/\Delta-1}\sum_{v\in V}\frac{1}{M}\mathbb{E}[\mu_v].$$

Because

$$
\begin{aligned}
\lim_{M\to\infty} \mathbb{E}[\mu_v] &= \lim_{M\to\infty} (M-1)\left[1 - \sum_{d=0}^{\infty} p_d \left(1 - \frac{1}{M}\right)^d\right] \\
&= \lim_{M\to\infty} (M-1)\left[1 - \sum_{d=0}^{\infty} p_d \left(1 - \frac{d}{M}\right)\right] \\
&= \lim_{M\to\infty} \frac{M-1}{M} \langle D\rangle = \langle D\rangle,
\end{aligned}
$$

we know

$$
\lim_{M\to\infty} \mathbb{E}\mathcal{C}_2 = \mathcal{Q}\frac{H|V|}{\Delta M}\langle D\rangle.
$$

Then

$$
\begin{aligned}
\lim_{M\to\infty} \mathbb{E}\mathcal{C} &= \lim_{M\to\infty}\left(\mathbb{E}\mathcal{C}_1 + \mathbb{E}\mathcal{C}_2\right) \\
&\le 4\log\left(\frac{|V|}{M}\right)\left(H\langle D\rangle\Lambda^* + \frac{H|V|}{\Delta M}\langle D\rangle\right) \\
&\le 4H\frac{|V|}{M}\log\left(\frac{|V|}{M}\right)\left[\frac{(\beta\langle D\rangle+\gamma)^2}{4\beta} + \frac{\langle D\rangle}{\Delta}\right].
\end{aligned}
$$

Theorem 2 exhibits good scalability in terms of $M$. If we consider the case where $M = 1$, the result in theorem 2 is correct if we remove the $\frac{\langle D\rangle}{\Delta}$ term.

**Definition 2**. Let random variable $\mathcal{M}$ be the total number of messages sent from all processes in one simulation instance.

**Theorem 3**.

$$
\mathcal{M} \le \frac{H}{\Delta}M(M-1).
$$

*Proof.* The proof is trivial since each process sends at most $M-1$ messages in an epoch.

## 4.2 Analysis of the dynamical system

In this section, we study the dynamical system that algorithm 5 in S1 File gives rise to. We study two extreme cases of this algorithm: $M = 1$, where all vertices belong to the same process, and $M = |V|$, where each vertex belongs to its own process.

Again we list our assumptions first:

- The state of a vertex $v$ can be modeled by a stochastic process, $S_v(t)$. For any given $t \in [0, H]$, $v \in V$, $S_v(t)$ is a Bernoulli random variable with

$$
\begin{aligned}
\Pr\{S_v(t) = 1\} &= \Pr\{v.\text{state@}t = \text{INF}\} = P_v(t), \\
\Pr\{S_v(t) = 0\} &= \Pr\{v.\text{state@}t = \text{SUS}\} = 1 - P_v(t).
\end{aligned}
$$

Again, we assume that $S_v(t)$ is independent of its neighborhood.

- The graph $G = (V, E)$ is a given, fixed graph represented by adjacency matrix $A \in \mathbb{R}^{|V|\times|V|}$. $A_{uv} = 1$ if $(u, v) \in E$ and 0 otherwise.

**Theorem 4**. *When M = 1, the* dynamics of algorithm 5 in S1 File *obey the master equation*

$$\frac{\partial P_v(t)}{\partial t} = \beta(1 - P_v(t)) \sum_{u \in V} A_{uv} P_u(t) - \gamma P_v(t). \quad (6)$$

*Proof*. When $M = 1$, the algorithm is reduced to the Next Reaction Method, which is faithful according to [14]. Note that the exact master equation would involve $2^{|V|}$ equations, which is intractable. By invoking the assumption on $v$'s independence of its neighborhood, we factorize the exact but cumbersome master equation into $|V|$ equations in the form of Eq (6), a technique also introduced in [47, 53, 54]. This technique is called by many in the field as "one-vertex quenched mean-field theory".

Consider $M = |V|$. During the $l$-th epoch $[(l − 1)\Delta, l\Delta]$, a vertex $v$ can only rely on a snapshot of its neighbors' states taken at time $(l − 1)\Delta$. We define the total infection rate from its neighbors $\beta_v^l$ in the $l$-th epoch as

$$\beta_v^l = \beta \sum_{u \in V} A_{uv} P_u((l - 1)\Delta).$$

The state of vertex $v$ during the $l$-th epoch follows the continuous-time Markov chain (CTMC) in Fig 3.

The CTMC in Fig 3 has rate matrix

$$\mathbf{Q}_v^l = \begin{bmatrix} -\gamma & \gamma \\ \beta_v^l & -\beta_v^l \end{bmatrix},$$

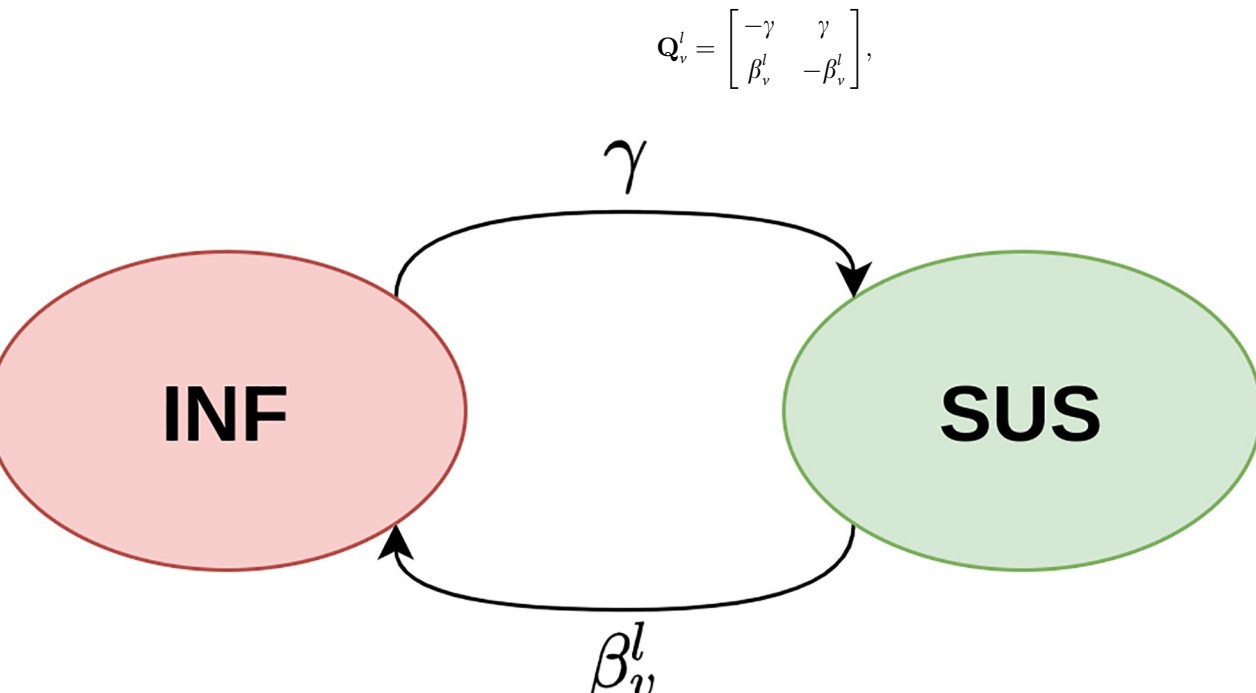

**Fig 3. The CTMC that the state of vertex $v$ follows during the $l$-th epoch.** Transition rates $\gamma$ and $\beta_v^l$ are constant during the $l$-th epoch.

which gives the transition matrix

$$\mathbf{P}_v^l(t) = e^{t\mathbf{Q}_v^l} = \begin{bmatrix} \dfrac{\beta_v^l}{\gamma + \beta_v^l} + \dfrac{\gamma \exp\{-t(\gamma + \beta_v^l)\}}{\gamma + \beta_v^l} & \dfrac{\gamma}{\gamma + \beta_v^l} - \dfrac{\gamma \exp\{-t(\gamma + \beta_v^l)\}}{\gamma + \beta_v^l} \\[4mm] \dfrac{\beta_v^l}{\gamma + \beta_v^l} - \dfrac{\beta_v^l \exp\{-t(\gamma + \beta_v^l)\}}{\gamma + \beta_v^l} & \dfrac{\gamma}{\gamma + \beta_v^l} + \dfrac{\beta_v^l \exp\{-t(\gamma + \beta_v^l)\}}{\gamma + \beta_v^l} \end{bmatrix}. \quad (7)$$

To distinguish the dynamics between $M = 1$ and $M = |V|$, we use different symbols to denote the (marginal) probability of vertex $v$ being infected at time $t$. We use $P_v(t)$ for $M = 1$ and $Q_v(t)$ for $M = |V|$. Let us also adopt the following shorthand notation:

$$P_v^l = P_v(\Delta(l-1)), \; Q_v^l = Q_v(\Delta(l-1)), \quad \tilde{\beta}_v^l = \beta \sum_{u \in \mathcal{N}(v)} P_u^l, \; \beta_v^l = \beta \sum_{u \in \mathcal{N}(v)} Q_u^l.$$

We show in theorem 5 that the difference between the two dynamics is of the same order of magnitude as $\Delta$, meaning that choosing a smaller $\Delta$ leads to a smaller loss of faithfulness.

Let $\phi(\Delta)$ be any function that satisfies

$$\lim_{\Delta \to 0^+} \phi(\Delta) = 0,$$

while we **do not necessarily** have

$$\lim_{\Delta \to 0^+} \frac{\phi(\Delta)}{\Delta} = 0.$$

It is useful to think of $\phi(\Delta)$ as a set of functions instead of a particular function. It is the set of functions whose Taylor expansion around 0 has the form

$$a_1 \Delta + a_2 \Delta^2 + a_3 \Delta^3 + \dots$$

Obviously, this set of functions is closed under addition and scalar product, which is why we can abuse the notation by writing

$$a\phi(\Delta) + b\phi(\Delta) = \phi(\Delta).$$

Similarly, let $\mathcal{O}(\Delta)$ be any function of $\Delta$ that satisfies

$$\lim_{\Delta \to 0^+} \frac{\mathcal{O}(\Delta)}{\Delta} = 0.$$

Note that any function that is $\mathcal{O}(\Delta)$ is $\phi(\Delta)$, but not the other way around. It is the set of functions whose Taylor expansion around 0 has the form

$$a_1 \Delta^2 + a_2 \Delta^3 + a_3 \Delta^4 + \dots$$

**Theorem 5**. *Suppose we are given the same initial condition, that is, $\forall v \in V, P_v(0) = Q_v(0)$. For any $T \in (0, H)$ s.t. $L = T/\Delta \in \mathbb{N}$, for any $v \in V, P_v(T) - Q_v(T)$ is $\phi(\Delta)$.*

*Proof.* For $M = 1$, the dynamics can be described by the differential equation

$$\forall v \in V, \quad \frac{\partial P_v(t)}{\partial t} = \beta(1 - P_v(t)) \sum_{u \in \mathcal{N}(v)} P_u(t) - \gamma P_v(t),$$

with its integral form being

$$P_v(T) - P_v(0) = \int_0^T dt \left\{ \beta(1 - P_v(t)) \sum_{u \in \mathcal{N}(v)} P_u(t) - \gamma P_v(t) \right\}.$$

We can rewrite the integral as the sum of a residual term and the Riemann sum,

$$P_v(T) - P_v(0)$$

$$= \phi(\Delta) + \sum_{l=1}^{L} \Delta \left\{ \beta(1 - P_v((l-1)\Delta)) \sum_{u \in \mathcal{N}(v)} P_u((l-1)\Delta) - \gamma P_v((l-1)\Delta) \right\}$$

$$= \phi(\Delta) + \sum_{l=1}^{L} \Delta \left\{ \beta(1 - P_v^l) \sum_{u \in \mathcal{N}(v)} P_u^l - \gamma P_v^l \right\}$$

$$= \phi(\Delta) + \Delta \sum_{l=1}^{L} [(1 - P_v^l)\tilde{\beta}_v^l - \gamma P_v^l].$$

For $M = |V|$, the transition matrix between epochs is

$$\mathbf{P}_v^l(\Delta) = \begin{bmatrix} \dfrac{\beta_v^l}{\gamma + \beta_v^l} + \dfrac{\gamma \exp\{-\Delta(\gamma + \beta_v^l)\}}{\gamma + \beta_v^l} & \dfrac{\gamma}{\gamma + \beta_v^l} - \dfrac{\gamma \exp\{-\Delta(\gamma + \beta_v^l)\}}{\gamma + \beta_v^l} \\[4mm] \dfrac{\beta_v^l}{\gamma + \beta_v^l} - \dfrac{\beta_v^l \exp\{-\Delta(\gamma + \beta_v^l)\}}{\gamma + \beta_v^l} & \dfrac{\gamma}{\gamma + \beta_v^l} + \dfrac{\beta_v^l \exp\{-\Delta(\gamma + \beta_v^l)\}}{\gamma + \beta_v^l} \end{bmatrix}.$$

Let the probability mass of vertex $v$ be

$$\pi_v(t) = \begin{bmatrix} Q_v(t) \\ 1 - Q_v(t) \end{bmatrix}.$$

And similarly

$$\pi_v^l = \begin{bmatrix} Q_v^l \\ 1 - Q_v^l \end{bmatrix} = \begin{bmatrix} Q_v(\Delta(l-1)) \\ 1 - Q_v(\Delta(l-1)) \end{bmatrix}.$$

Thanks to the CTMC,

$$\pi_v^{l+1} = [\mathbf{P}_v^l(\Delta)]^{\mathsf{T}} \pi_v^l,$$

which implies that

$$Q_v^{l+1} = Q_v^l \left[ \frac{\beta_v^l + \gamma \exp\{-\Delta(\gamma + \beta_v^l)\}}{\gamma + \beta_v^l} \right] + \left(1 - Q_v^l\right) \left[ \frac{\beta_v^l - \beta_v^l \exp\{-\Delta(\gamma + \beta_v^l)\}}{\gamma + \beta_v^l} \right]. \qquad (8)$$

A simple Taylor expansion of Eq (8) gives us

$$Q_v^{l+1} - Q_v^l = \mathcal{O}(\Delta) + (1 - Q_v^l)\Delta\beta_v^l - Q_v^l\Delta\gamma.$$

A telescoping sum gives us

$$
\begin{aligned}
Q_v(T) - Q_v(0) \quad &= Q_v^{L+1} - Q_v^1 = \sum_{l=1}^{L}(Q_v^{l+1} - Q_v^l) \\
&= \left[\sum_{l=1}^{L}\mathcal{O}(\Delta)\right] + \left[\sum_{l=1}^{L}(1 - Q_v^l)\Delta\beta_v^l - Q_v^l\Delta\gamma\right] \\
&= \phi(\Delta) + \Delta\sum_{l=1}^{L}[(1 - Q_v^l)\beta_v^l - \gamma Q_v^l].
\end{aligned}
$$

Here, $\sum_{l=1}^{L}\mathcal{O}(\Delta)$ yields $\phi(\Delta)$ because

$$
\sum_{l=1}^{L}\mathcal{O}(\Delta) = \frac{T}{\Delta}\mathcal{O}(\Delta),
$$

which does go to zero when $\Delta \to 0$.

In fact, for any $\lambda = 0, 1, \ldots, L-1, L$,

$$
P_v^{\lambda+1} - P_v^1 = \phi(\Delta) + \Delta\sum_{l=1}^{\lambda}[(1 - P_v^l)\tilde{\beta}_v^l - \gamma P_v^l], \tag{9}
$$

$$
Q_v^{\lambda+1} - Q_v^1 = \phi(\Delta) + \Delta\sum_{l=1}^{\lambda}[(1 - Q_v^l)\beta_v^l - \gamma Q_v^l], \tag{10}
$$

using the same reasoning. Eqs (9) and (10) describe a recurrent relation which is conducive to a proof by induction. We want to show that

$$
\forall v \in V, \forall l \in \{1, 2, \ldots, L+1\}, P_v^l - Q_v^l \text{ is } \phi(\Delta).
$$

Let $0 < L' < L$. Let us assume for the sake of argument that $\forall v \in V, P_v^\lambda - Q_v^\lambda$ is $\phi(\Delta)$ for any $\lambda = 1, 2, \ldots, L'-1, L'$. The base case is straightforward with $P_v^1 - Q_v^1$ being zero (the same initial condition). For the inductive case, we need to show that

$$
\forall v \in V, \forall \lambda \in \{1, 2 \ldots, L'\}, P_v^\lambda - Q_v^\lambda \text{ is } \phi(\Delta) \Rightarrow \forall v \in V, P_v^{L'+1} - Q_v^{L'+1} \text{ is } \phi(\Delta).
$$

To show this, note that

$$
\begin{aligned}
P_v^{L'+1} - P_v^1 \quad &= \phi(\Delta) + \Delta\sum_{l=1}^{L'}[(1 - P_v^l)\tilde{\beta}_v^l - \gamma P_v^l], \\
Q_v^{L'+1} - Q_v^1 \quad &= \phi(\Delta) + \Delta\sum_{l=1}^{L'}[(1 - Q_v^l)\beta_v^l - \gamma Q_v^l].
\end{aligned}
$$

A subtraction gives us

$$P_v^{L'+1} - Q_v^{L'+1} = \phi(\Delta) + \Delta\sum_{l=1}^{L'}\Big\{[(1-P_v^l)\tilde{\beta}_v^l - (1-Q_v^l)\beta_v^l] - \gamma(P_v^l - Q_v^l)\Big\}$$

$$= \phi(\Delta) + \Delta\sum_{l=1}^{L'}\Big\{(1-P_v^l)(\tilde{\beta}_v^l - \beta_v^l) - (P_v^l - Q_v^l)\beta_v^l - \gamma(P_v^l - Q_v^l)\Big\}$$

$$= \phi(\Delta) + \sum_{l=1}^{L'}\Delta\phi(\Delta) = \phi(\Delta) + \sum_{l=1}^{L'}\mathcal{O}(\Delta) = \phi(\Delta),$$

which is exactly what we need. Note that $\tilde{\beta}_v^l - \beta_v^l$ is also $\phi(\Delta)$ because

$$\tilde{\beta}_v^l - \beta_v^l = \beta\sum_{u\in\mathcal{N}(v)}(P_u^l - Q_u^l) = \beta\sum_{u\in\mathcal{N}(v)}\phi(\Delta) = \phi(\Delta).$$

Also $\Delta\phi(\Delta)$ yields $\mathcal{O}(\Delta)$ because

$$\lim_{\Delta\to 0^+}\frac{\Delta\phi(\Delta)}{\Delta} = \lim_{\Delta\to 0^+}\phi(\Delta) = 0.$$

We know $\forall v\in V, \forall l\in\{1,2,\ldots,L+1\}, P_v^l - Q_v^l$ is $\phi(\Delta)$. In particular, $\forall v\in V, P_v^{L+1} - Q_v^{L+1} = P_v(T) - Q_v(T)$ is $\phi(\Delta)$.

**Definition 3**. Let the state vector $\vec{\rho}(t)\in[0,1]^{|V|}$ be

$$\vec{\rho}(t) = \begin{bmatrix} P_{v_1}(t) & P_{v_2}(t) & \ldots \end{bmatrix}_{v_i\in V}^\mathsf{T}.$$

This definition makes sense for both $M=1$ and $M=|V|$. We say $\beta^*$ is the epidemic threshold if, all other input parameters being fixed,

- when $\beta < \beta^*$, $\vec{\rho} = 0$ is a locally asymptotically stable fixed point;

- when $\beta > \beta^*$, $\vec{\rho} = 0$ is an unstable fixed point.

We let $\lambda_1^A$ be the largest eigenvalue of the adjacency matrix $A$. Because we only discuss the dynamics of $M=|V|$, we simply use $P_v(t)$ for $M=|V|$. We use the shorthand

$$P_v^l = P_v(\Delta(l-1)), \quad \vec{\rho}_l = \begin{bmatrix} P_{v_1}^l & P_{v_2}^l & \ldots \end{bmatrix}_{v_i\in V}^\mathsf{T}.$$

**Theorem 6**. *When $M=|V|$, as $\Delta\to\infty$, the dynamics of the algorithm exhibit the same epidemic threshold as the dynamics in* Eq (6).

*Proof*. First, we have

$$\lim_{\Delta\to\infty}\mathbf{P}_v^l(\Delta) = \lim_{\Delta\to\infty}\begin{bmatrix} \dfrac{\beta_v^l}{\gamma+\beta_v^l} & \dfrac{\gamma}{\gamma+\beta_v^l} \\ \dfrac{\beta_v^l}{\gamma+\beta_v^l} & \dfrac{\gamma}{\gamma+\beta_v^l} \end{bmatrix}.$$

Indeed, as $\Delta\to\infty$, we recover the stationary distribution, and vertex $v$ essentially forgets about its state at the beginning of epoch $l$, which allows us to write

$$P_v^l = \frac{\beta\sum_{u\in V}A_{uv}P_u^{l-1}}{\gamma+\beta\sum_{u\in V}A_{uv}P_u^{l-1}}. \tag{11}$$

We have obtained a discrete autonomous system about $\vec{\rho_l}$. We see that $\vec{\rho_l} = 0$ is a fixed point. Let us linearize the discrete system in Eq (11) by computing the Jacobian,

$$J_{uv} = \frac{\partial P_u^l}{\partial P_v^{l-1}} = \beta A_{uv} \frac{\partial P_u^l}{\partial \beta_u^l} = \beta A_{uv} \frac{\gamma}{\left(\gamma + \beta_v^l\right)^2},$$

meaning that if we let $J_0 = J|_{\vec{\rho}=0}$, and let $\lambda_1^{J_0}$ be the largest eigenvalue of $J_0$,

$$J_0 = \frac{\beta}{\gamma} A, \quad \lambda_1^{J_0} = \frac{\beta}{\gamma} \lambda_1^A,$$

which indicates that $\rho = 0$ is locally asymptotically stable if and only if

$$\beta < \frac{\gamma}{\lambda_1^A} = \beta^*,$$

and unstable if and only if

$$\beta > \frac{\gamma}{\lambda_1^A} = \beta^*.$$

This result is in line with the results in [55], proving the claim. A simple linear expansion on the right-hand side of Eq (6) yields the same conclusion.

**Theorem 7**. *The system in Eq (6) and the system in Eq (11) have the same fixed points.*

*Proof.* Simply set the right-hand side of Eq (6) to zero, yielding

$$P_v = \frac{\beta \sum_{u \in V} A_{uv} P_u}{\gamma + \beta \sum_{u \in V} A_{uv} P_u}. \tag{12}$$

Set $\forall v \in V, P_v^l = P_v^{l-1} = \pi_v$ in Eq (11), yielding

$$\pi_v = \frac{\beta \sum_{u \in V} A_{uv} \pi_u}{\gamma + \beta \sum_{u \in V} A_{uv} \pi_u}. \tag{13}$$

Eqs (12) and (13) are equivalent and must result in the same set of fixed points.

Theorem 5 is reassuring in the sense that even if the graph is partitioned to the utmost with $M = |V|$, a smaller $\Delta$ does lead to smaller loss of faithfulness.

Meanwhile, theorems 6 and 7 deals with a highly adversarial scenario: $M = |V|$ and $\Delta \to \infty$. Intuitively, the longer $\Delta$ is, the more stale snapshots of neighbor states will be. Also, the more parts there are in the partition, the more commonly we need to rely on stale neighbor states. Theorems 6 and 7 tell us that even under this challenging scenario, algorithm 5 in S1 File can still recover some of the most distinctive features of the dynamical system that it is supposed to simulate.

Note that even though algorithm 5 in S1 File is equivalent to the Next Reaction Method [14] commonly used in EDS when $M = 1$, it is not equivalent to TDS when $M = |V|$, at least not with the popular method of rejection sampling used among practitioners [1, 2]. The intuition is simple: in a scheme of rejection sampling and synchronous update, the state of a vertex can change at most once in an epoch, ruling out any possibility of re-infection, which may have a nontrivial impact on the trajectory, especially with large $\Delta$. Indeed, rejection sampling is simply a linear approximation of the transition matrix in Eq (7). If we have a variant of TDS that employs the exact transition matrix in Eq (7), then an equivalence can be established between this variant of TDS and algorithm 5 in S1 File.

## 5 Experimental results

In this section, we design experiments to demonstrate the faithfulness and scalability (time consumption and performance breakdown) of algorithm 5 in S1 File. The MPI implementation we choose is MPICH [56].

For all experiments in this section, we collect the turnaround time $T_i^r$ on process $P_i$ in the $r$-th repeated experiment for all $i \in \{1, 2, \ldots, M\}$ and $r \in \{1, 2, \ldots, R\}$. We compute the total time consumption $\bar{T}$ as

$$\bar{T} = \frac{1}{RM} \sum_{r=1}^{R} \sum_{i=1}^{M} T_i^r.$$

The time spent blocking is computed in the same way.

As for average trajectories, we collect infection count $C_l^r$ at time $l\Delta$ in the $r$-th repeated experiment. $l \in \left\{0, 1, 2, \ldots, \frac{H}{\Delta}\right\}$. $r \in \{1, 2, \ldots, R\}$. We do not collect the entire trajectory, which involves merging event logs on all processes and visualizing data at the scale of gigabytes. Instead, we pick integer multiples of $\Delta$ and compute averages as

$$\bar{C}_l = \frac{1}{R} \sum_{r=1}^{R} C_l^r.$$

### 5.1 An SIS epidemic on an Erdos-Renyi graph

In this section, we run our experiments on one single machine, using the multiprocessing feature of MPI [38]. This setup limits the scale of experiments we can handle (memory limits), but it ensures low-latency communication among processes.

We synthesize a standard $G(N, p)$ Erdos-Renyi graph [57] with number of vertices $N = 100000$ and edge probability $p = 10^{-4}$, which we refer to as *er100k*. We select uniformly at random 1% of vertices to be initially infected. This graph is used for all repeated runs. $\gamma = 0.25$. $\beta = 0.05$. $H = 60$.

We run experiments on the following grid of parameter settings: $M = 1, 2, 4, 8$ and $\Delta = 0.1, 0.2, \ldots, 0.9, 1.0$. For each choice of parameter $M$, the graph is evenly partitioned into $M$ parts. The partitioning is independent of vertex states and graph connectivity. The partitioning is fixed for all experiments with the same $M$. We run 20 repeated experiments to take averages for each point in the parameter grid.

Besides $M = 1, 2, 4, 8$, we also implement TDS with simple rejection sampling.

We plot the average trajectories in Fig 4 to assess the faithfulness of the algorithm under various parameter settings. We also plot in Fig 5 the relative divergence from the setup with $M = 1$, where the algorithm is reduced to a standard EDS.

We collect timing information in this experiment, shown in Fig 6.

### 5.2 Stochastic block model

In this section, we study the relationship between graph connectivity and algorithmic scalability. We prepare $N = 100000$ vertices with 40% of them being initially infected (selected uniformly at random). We partition these vertices evenly into eight blocks and assign them to $M = 8$ processes accordingly. The partitioning is independent of vertex states. The size of each block is $n = N/M = 12500$.

The graph connectivity obeys the stochastic block model (SBM) [58]. We fix the mean degree of all synthesized SBMs to be $\langle D \rangle = 10$. We refer to the probability of finding an edge

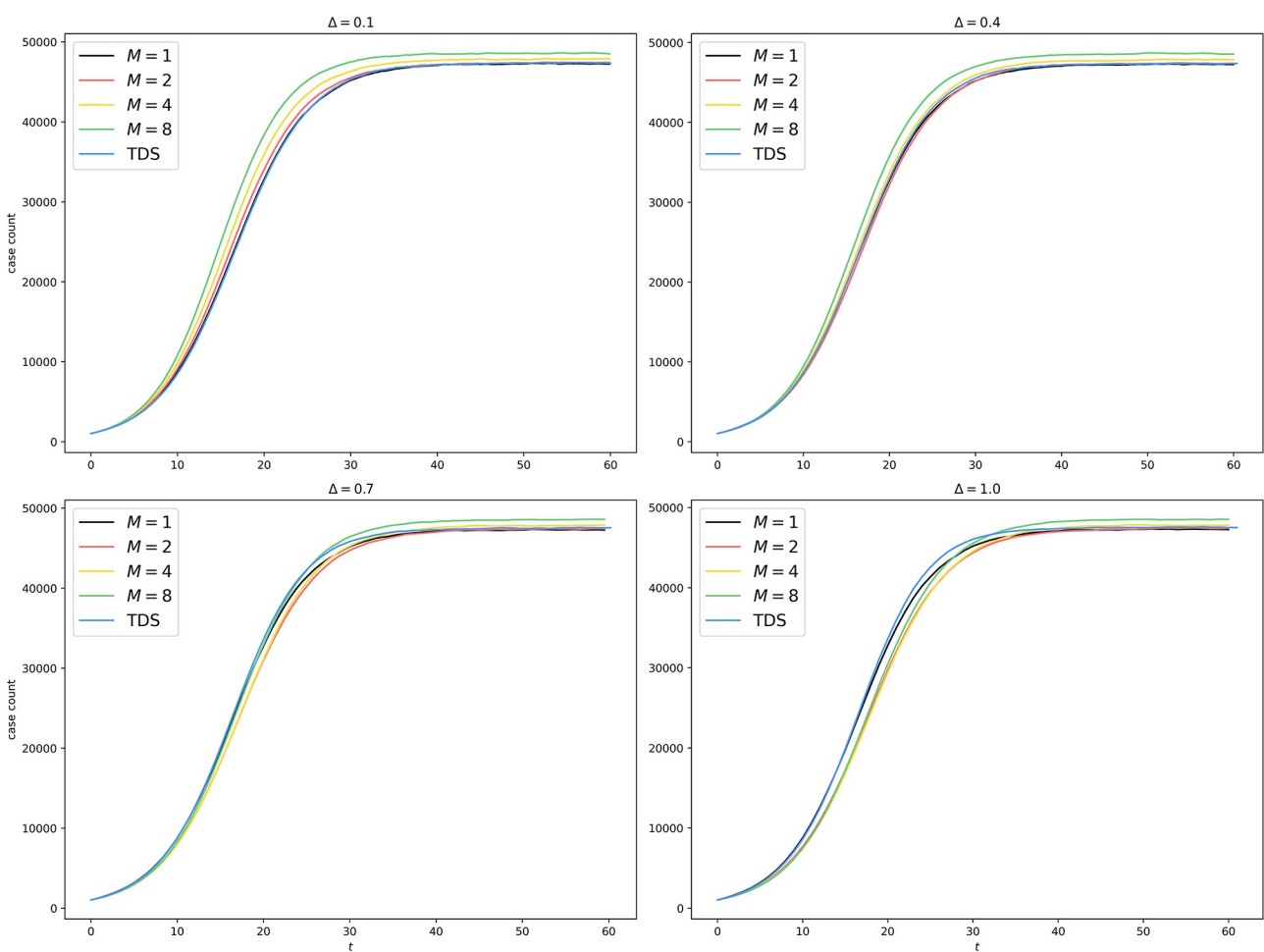

**Fig 4. Uniform partitioning of er100k: Average trajectories with different $M$ and $\Delta$.**

between two vertices from the same block as $p_i$ and the probability of finding an edge between two vertices from different blocks as $p_o$. We have

$$\langle D \rangle = p_i(n-1) + p_o(N-n).$$

Once we fix $p_o$, $p_i$ can be computed as

$$p_i = \frac{1}{n-1}[\langle D \rangle - p_o(N-n)].$$

We synthesize 20 SBMs with parameters shown in Table 1.

For each $(p_i, p_o)$ pair in Table 1, we generate a stochastic block model, on which we conduct 20 repeated runs of algorithm 5 in S1 File with $M = 1$ as the baseline and 20 repeated runs with $M = 8$ to compute the speed-up factor. The results are shown in Fig 7.

### 5.3 Experiments with a real graph on an MPI cluster

In this section, we run our experiments on a cluster of four machines using the same MPI implementation as before. Each of the four machines has the following hardware setup: 2x12cores@2.5 GHz, 256GB RAM, 2x240GB SSD, 2x2TB@7200RPM.

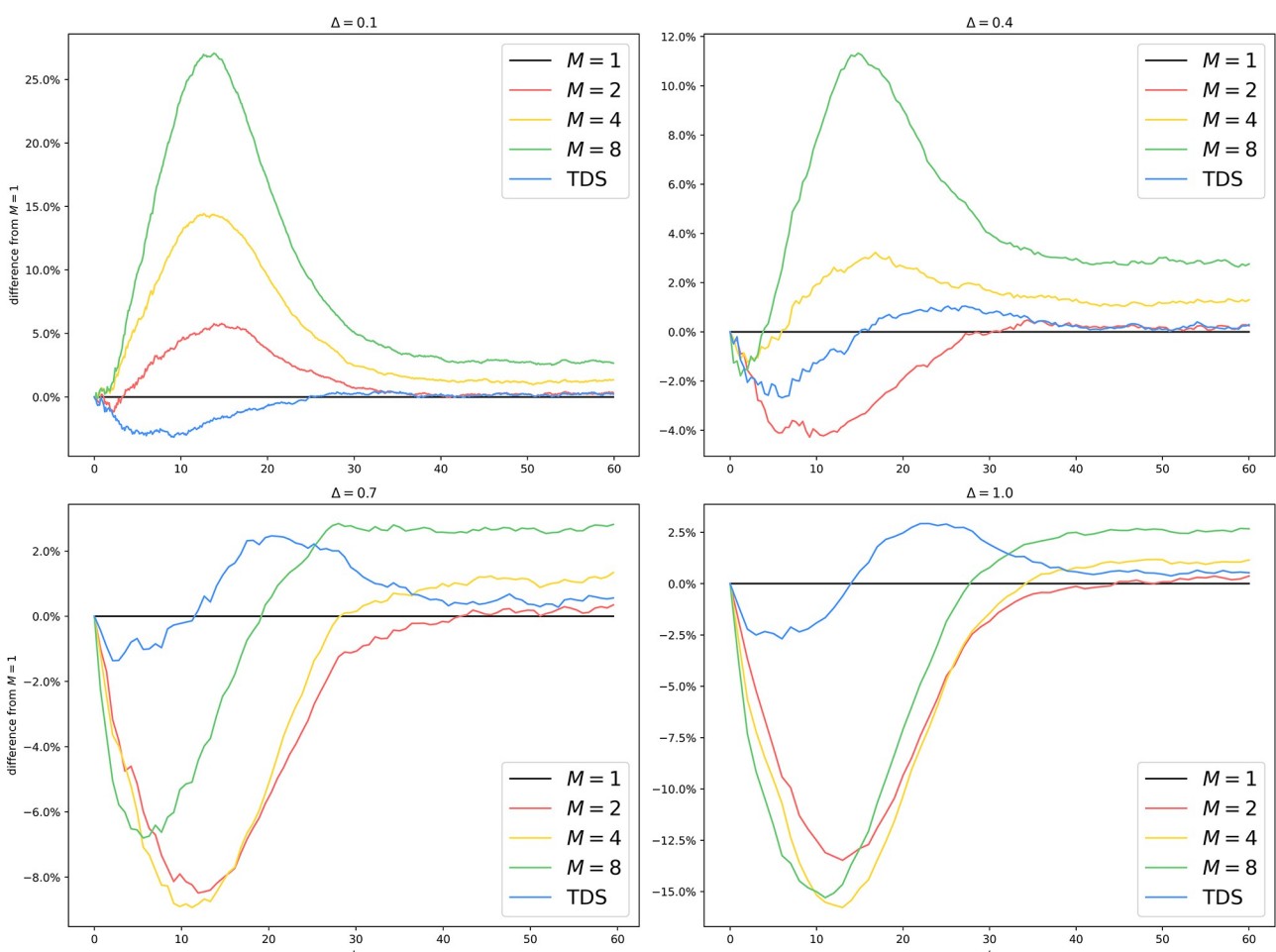

**Fig 5. Uniform partitioning of er100k: Difference from $M = 1$ with different $M$ and $\Delta$.** The horizontal line associated with $M = 1$ represents the ground truth. All other curves show some lack of faithfulness. When $\Delta$ is small ($\Delta = 0.1, 0.4$), $M > 1$ tends to overestimate the spread of the epidemic since by making a mean-field approximation on the border, we permit interprocess infection events like $\text{Inf}(Q \rightarrow v)$ to occur even if the border vertex $v$ has no infected neighbor on $Q$ at all. When $\Delta$ is large ($\Delta = 0.7, 1.0$), $M > 1$ tends to underestimate the spread of the epidemic since the more outdated the information about other processes is, the more this process underestimates the border infection probability throughout the epoch. The main divergence happens during the exponential growth phase, where a small delay in time causes significant differences in case count. Once the epidemic reaches equilibrium, the differences are diminished ($< 5\%$) for all algorithms.

We choose a graph from a real-world blogging community called LiveJournal, where people declare friendship with each other, forming an undirected graph with one connected component of 3997962 vertices and 34681189 edges [59]. We refer to this graph as *lj4m*.

We use METIS [32, 51, 60] for graph partitioning. METIS partitions graphs so that all parts have roughly the same size and edges between different parts are few. We partition the Live-Journal graph into 12, 24, 48, and 96 parts, using the default parameter setting of METIS [32]. We assign the parts evenly to the four machines connected by LAN (Ethernet), each having 3, 6, 12, and 24 parts. A maximum of 24 is chosen here because each machine has 24 cores.

0.5% of vertices are initialized to be infected, chosen uniformly at random, independent from graph partitioning. $\beta = 0.02$, $\gamma = 0.3$. $H = 60$. To take averages, we run 10 experiments for each parameter setting of $(M, \Delta)$ pair.

We collect average trajectories, shown in Fig 8. We also plot the relative divergence from the setup with $M = 1$ in Fig 9.

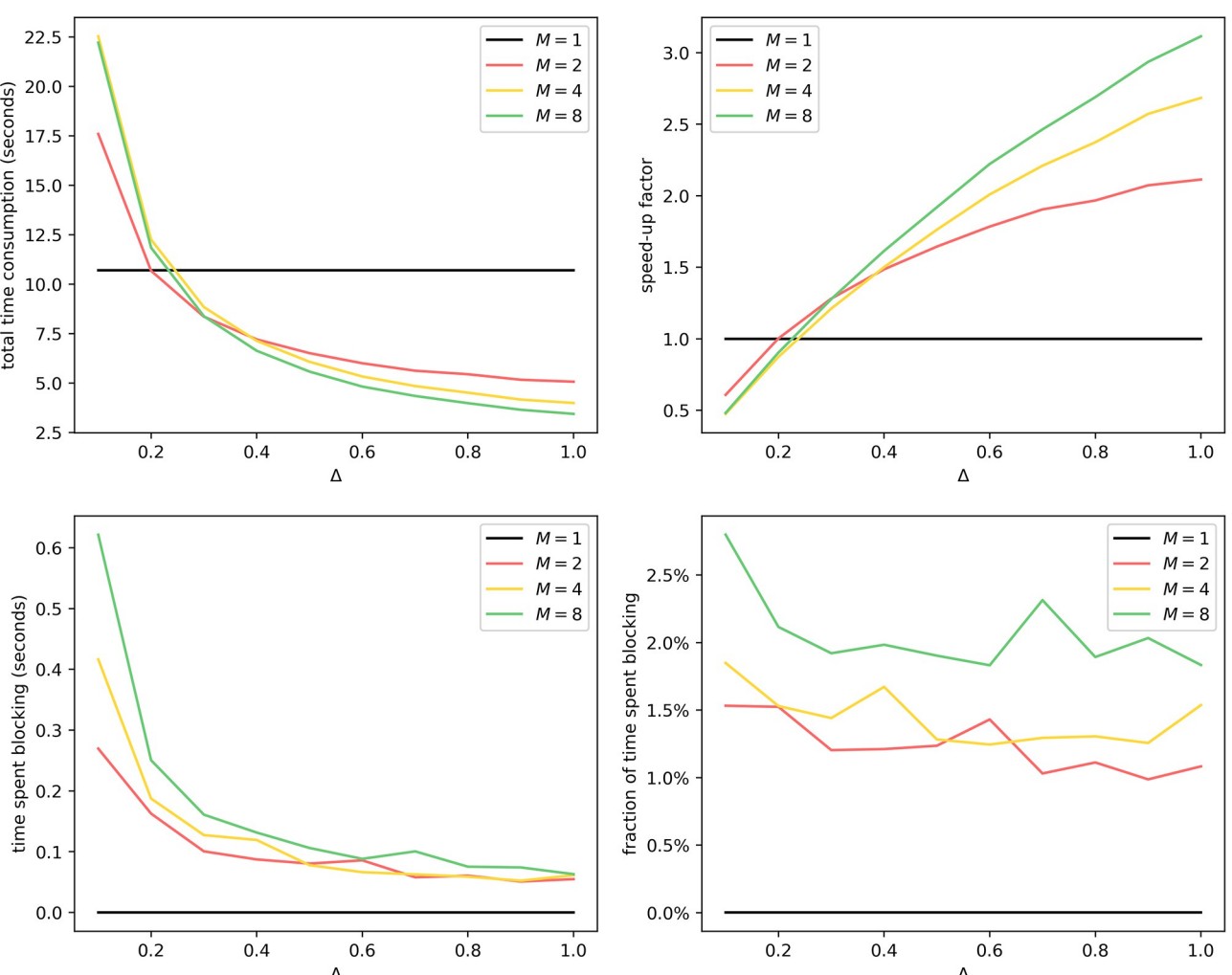

**Fig 6. Uniform partitioning of er100k: Average time consumption with different $M$ and $\Delta$.** For total time consumption, the break-even point is around $\Delta = 0.2$, since a smaller $\Delta$ causes more updates of interprocess infection events of the form $\text{Inf}(Q \rightarrow v)$. This is aligned with our result from theorem 2, since for $M = 1$, the overhead term $\langle D \rangle / \Delta$ is absent. Generally, total time consumption decreases as $M$ increases and as $\Delta$ increases. We have also shown that the time spent blocking grows as $M$ increases and shrinks as $\Delta$ increases, even though it accounts for less than 5% of total time consumption.

**Table 1. Parameters for SBMs.**

| $p_i$ | 8.0006e-4 | 7.5796e-4 | 7.1585e-4 | 6.7374e-4 | 6.3163e-4 |
|---|---|---|---|---|---|
| $p_o$ | 0.0000e+0 | 6.0151e-6 | 1.2030e-5 | 1.8045e-5 | 2.4060e-5 |
| $p_i$ | 5.8952e-4 | 5.4741e-4 | 5.0530e-4 | 4.6320e-4 | 4.2109e-4 |
| $p_o$ | 3.0075e-5 | 3.6090e-5 | 4.2105e-5 | 4.8120e-5 | 5.4135e-5 |
| $p_i$ | 3.7898e-4 | 3.3687e-4 | 2.9476e-4 | 2.5265e-4 | 2.1054e-4 |
| $p_o$ | 6.0150e-5 | 6.6165e-5 | 7.2181e-5 | 7.8195e-5 | 8.4211e-5 |
| $p_i$ | 1.6844e-4 | 1.2633e-4 | 8.4217e-5 | 4.2109e-5 | 0.0000e+0 |
| $p_o$ | 9.0225e-5 | 9.6241e-5 | 1.0226e-4 | 1.0827e-4 | 1.1429e-4 |

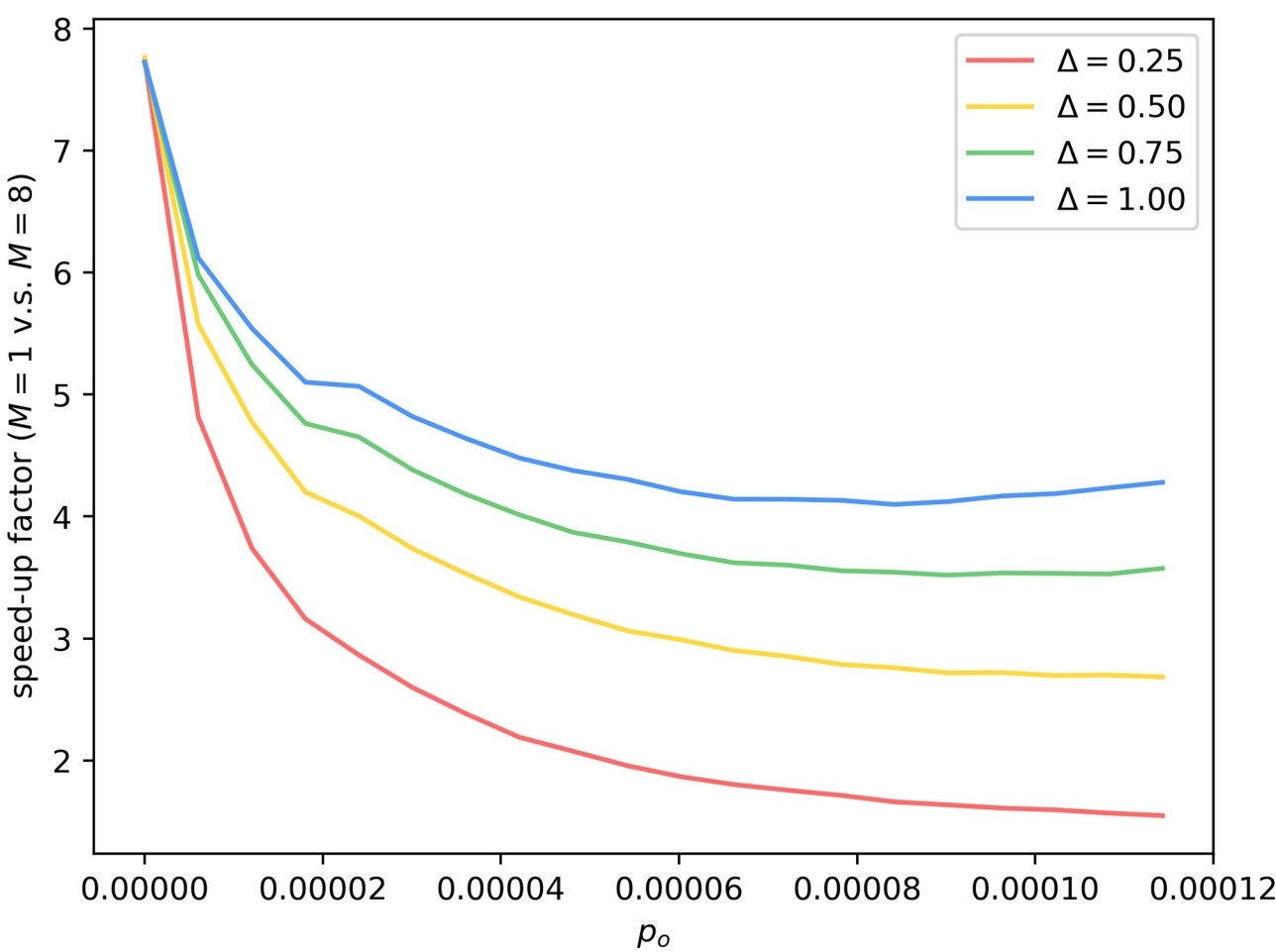

**Fig 7. Speed-up factor of $M = 8$ over $M = 1$.** Speed-up factor is almost 8 with $p_o = 0$, where the SBM is reduced to 8 disconnected components, and the task becomes embarrassingly parallel. In general, the speed-up factor goes up as $\Delta$ goes up and as $p_o$ goes down, both reducing the amount of communication required.

We also collect timing information, shown in Fig 10.

### 5.4 Ghost-cell implementation

In this section, we discuss a variant of algorithm 5 in S1 File, referred to as the "ghost-cell implementation", which demystifies the phenomenon observed in Figs 5 and 9 where $M > 1$ leads to a different equilibrium from that of $M = 1$.

First and foremost, lack of faithfulness occurs in two different types in Fig 5 with large $\Delta$ and Fig 9. The first type is about growth, where $M > 1$ underestimates the infection count during the exponential growth phase of the epidemic. The second type is about the equilibrium, where $M = 1$ and $M > 1$ eventually reach different equilibria. We are more concerned about the second type of error because:

1. the second type is harder to explain intuitively, while the first type can be explained by stale border states causing the lag;

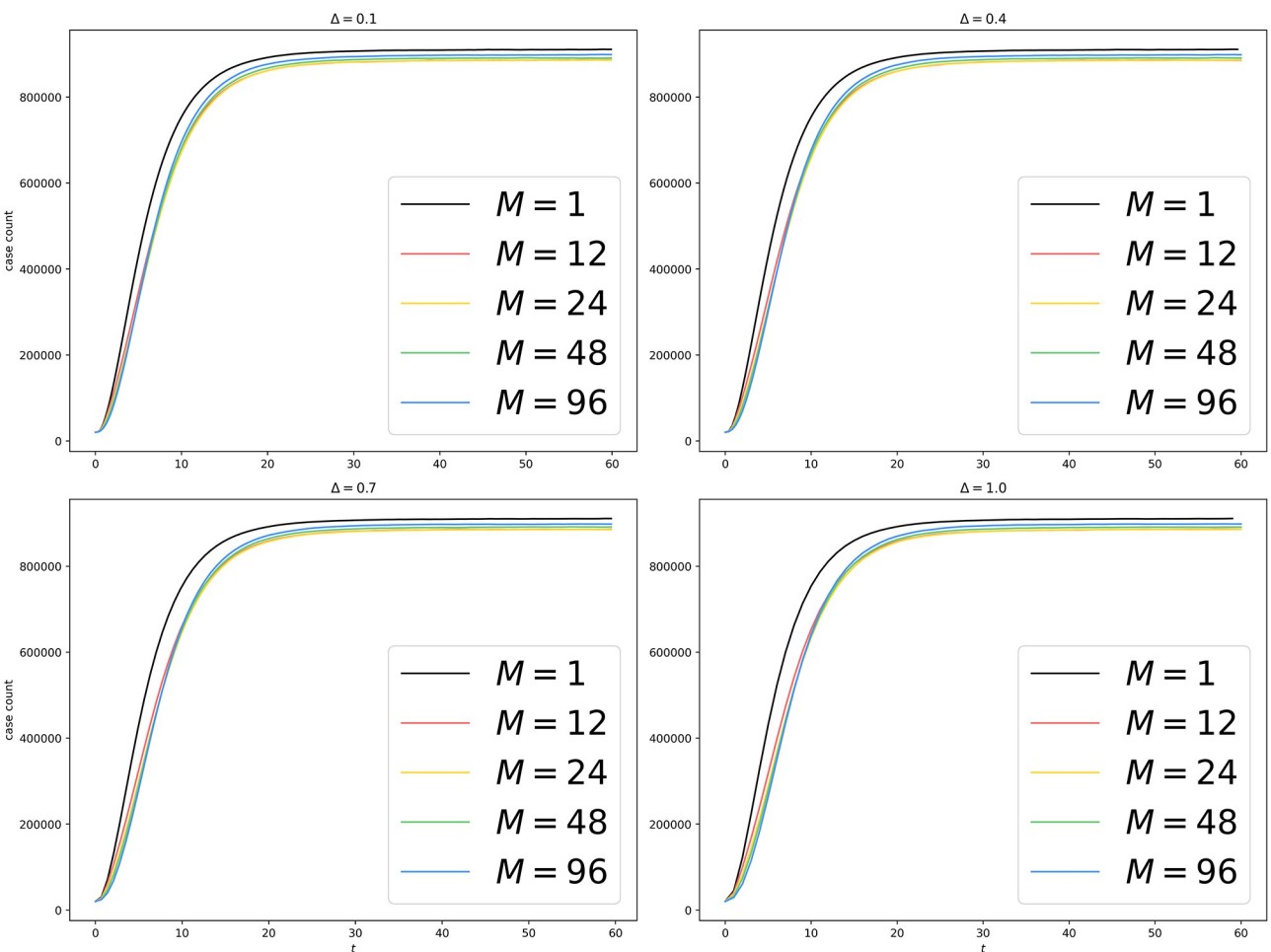

**Fig 8. METIS partitioning of lj4m: Average trajectories for an SIS epidemic on a graph of around 4 million vertices.**

2. the second type is more detrimental to practitioners since many are only interested in the epidemic size at equilibrium;

3. the existence of the second type contradicts theorem 7.

Because in theorem 7, we assume that $M = |V|$, meaning that every vertex is on its own process and no averaging is needed, we hypothesize that the border averaging trick causes incorrect equilibrium. To test our hypothesis, we modify algorithm 5 in S1 File to send the state of the entire border instead of a summary statistic. We refer to this variant of algorithm 5 in S1 File as the "*ghost-cell implementation*" (**GCI**).

We also use a new graph dataset: the DBLP collaboration network [59]. The dataset comes with ground truth communities,

$$C_1, C_2, \ldots C_K, \text{ with } C_k = \{v_{k_1}, v_{k_2}, \ldots\}.$$

These communities are not disjoint, nor do they cover all of $V$. We prune the graph by taking its subgraph induced by

$$V \cap (\cup_{k=1}^{K} C_k),$$

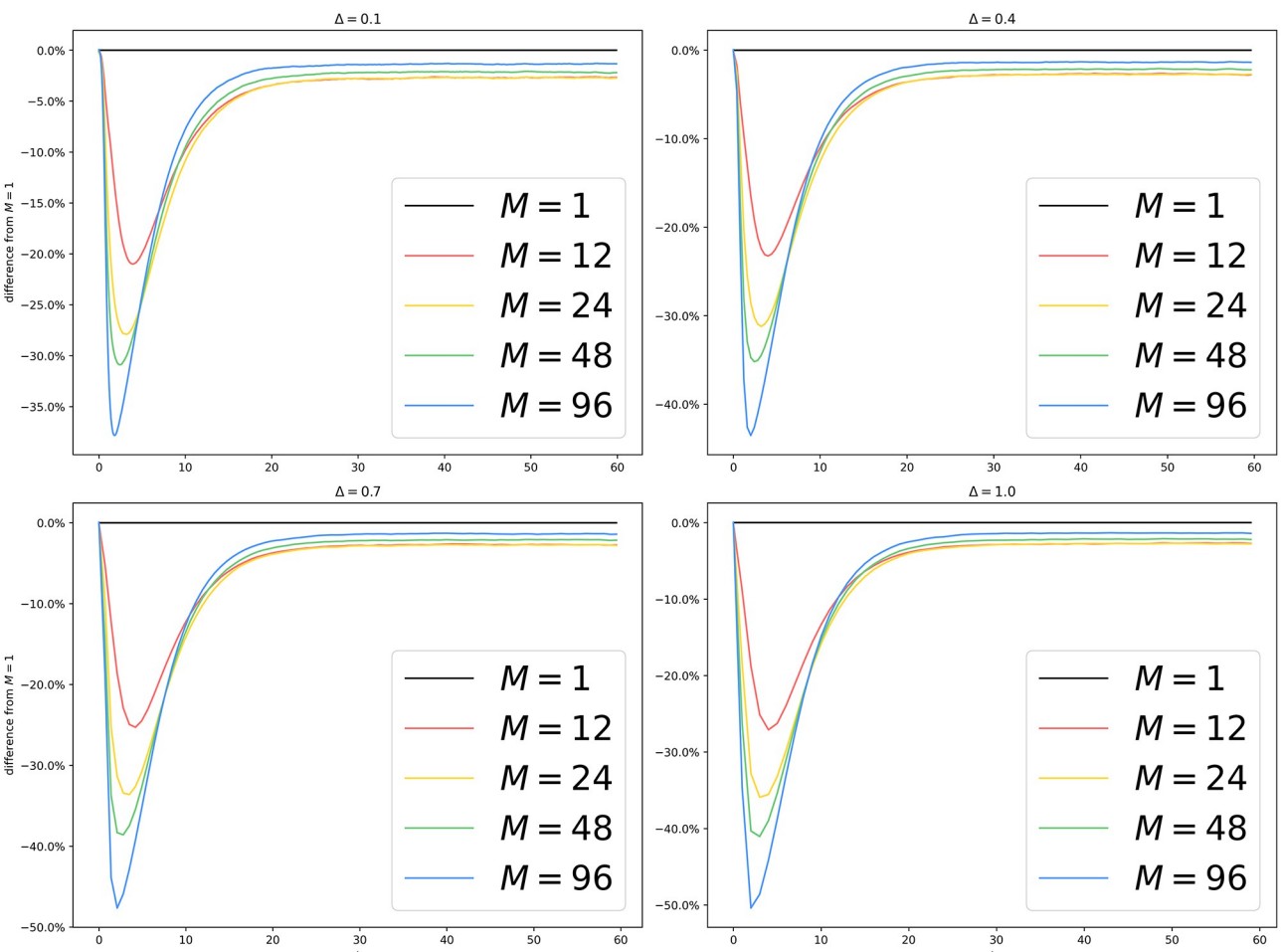

**Fig 9. METIS partitioning of lj4m: Difference from *M* = 1 with different *M* and Δ.** We see that, unlike Fig 5, all experiments with $M > 1$ underestimate the infection count. There are two potential causes. Firstly, the choice of Δ is way higher than the typical interevent time in the simulated system, which tends to shrink as the size of the system grows. Secondly, our mean-field approximation on the border tends to inflate the number of infected vertices, as we have discussed in Fig 5. This inflating effect is counterbalanced by our usage of the METIS graph partitioning library, which vastly reduces the number of interprocess edges and gives rise to smaller borders on all processes. Again, as the epidemic reaches equilibrium, the divergence shrinks to less than 5%.

yielding a subgraph with around 260k vertices, which we call *dblp260k*. We conduct this pre-processing to strengthen the community structure of the graph.

We run GCI on er100k and dblp260k, with the same parameters as in Fig 4. The results are shown in Figs 11 to 13.

## 6 Discussions

We have proposed in Section 2 and Section 3 an algorithm to simulate epidemics at a large scale with the help of parallel/distributed computing hardware. We have offered a well-tailored implementation in Section 7. We have analyzed its complexity in Section 4.1 and established that it scales well with the number of cores used while sending a modest number of messages. We have also studied in Section 4.2 its induced dynamical system, discovering that when each vertex is on its own process, the bias our algorithm introduces tends to zero as Δ tends to zero,

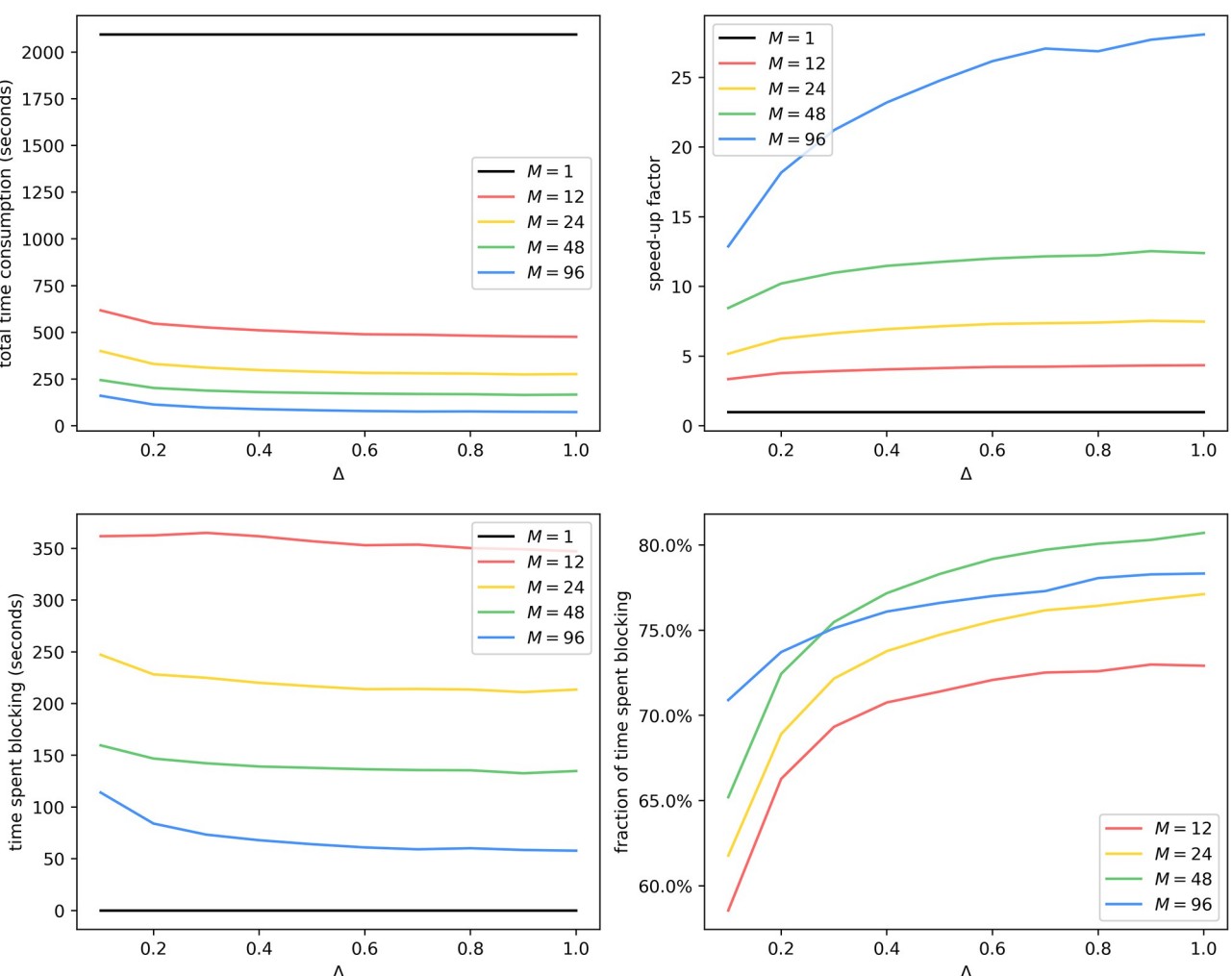

**Fig 10. METIS partitioning of lj4m: Average time consumption with different *M* and Δ.** Like Fig 6, we see a decrease in total time consumption as *M* increases and as Δ increases. Unlike Fig 6, time spent blocking goes down as *M* increases. Also unlike Fig 6, the fraction of time spent blocking goes up as Δ increases. We hypothesize that this is caused by poor load balancing because with a large Δ and a small *M*, the computational workload per epoch increases, meaning that a straggler process can lag behind more, keeping all other processes blocked. Meanwhile, simulations on er100k enjoy good load balancing because the graph and the partitioning scheme in Fig 6 are highly homogeneous.

and our algorithm preserves features interesting to epidemiologists such as the epidemic threshold even as Δ tends to $+\infty$. We find in Section 5.1 and Section 5.3 that the total time consumption goes down as *M* and Δ increase, however at the cost of faithfulness, as shown in Fig 4. We find in Section 5.2 that by reducing graph connectivity between parts/processes, we achieve better scalability, inspiring our decision to employ a graph partitioning software in Section 5.3. In Section 5.3, however, we see that a graph partitioning scheme can potentially be a double-edged sword in that it may worsen load balancing, as shown in Fig 16 in S1 File. Even though we have shown that poor load balancing can be the culprit, in our future work, we will investigate the true cause of the difference between Figs 6 and 10.

As for the limitations and extensions of our algorithm, let us review the scope of the problem.

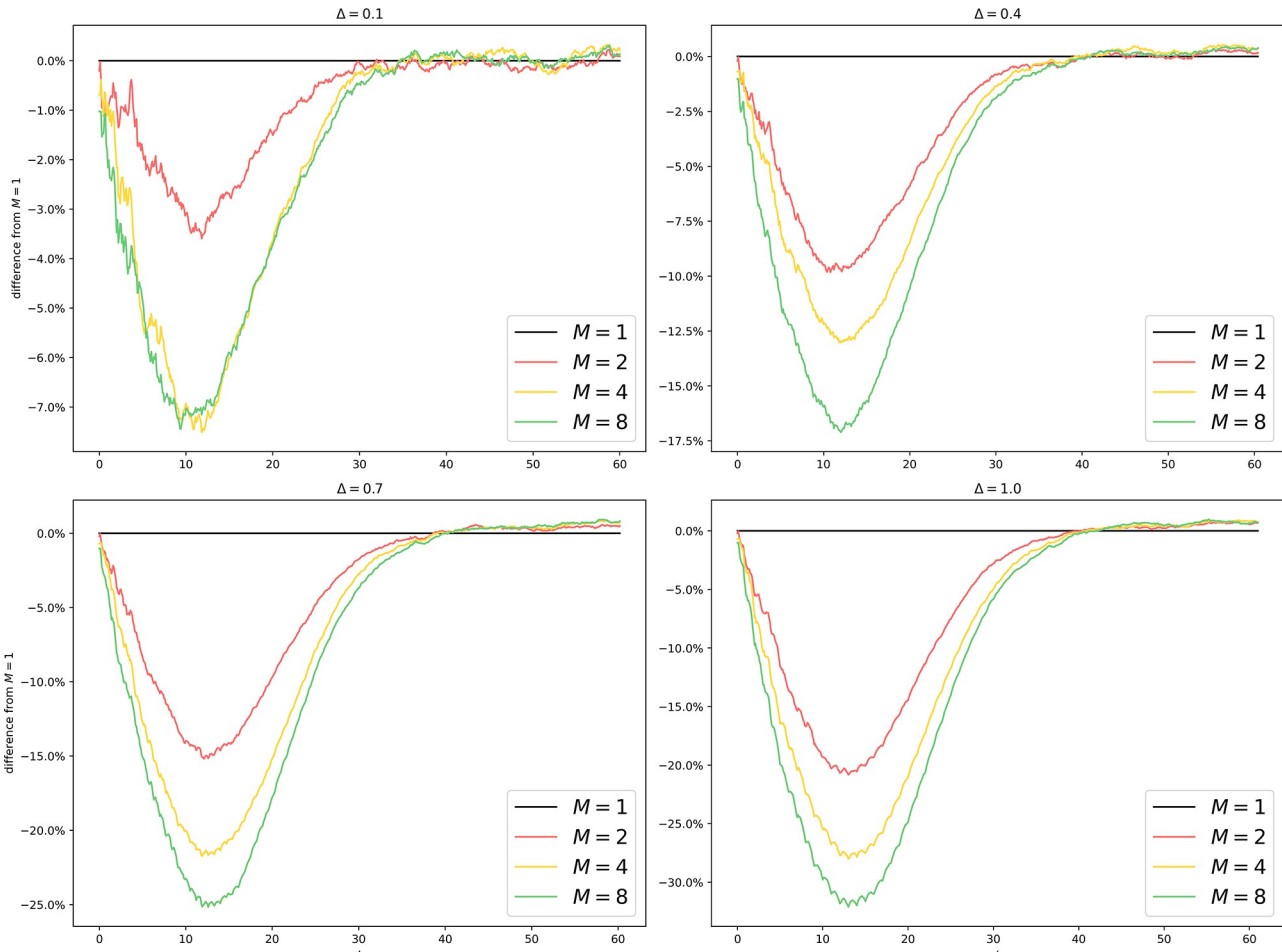

**Fig 11. Uniform partitioning of er100k (GCI): Difference from _M_ = 1 with different _M_ and Δ.** Compared to Fig 5, the lag (type-1) is still around but the difference in equilibria (type-2) has been reduced to less than 1%.

- "Markovian": Instead of rates of exponential distributions, we compute instantaneous rates as

$$\lambda(t) = \frac{\psi(t)}{1 - \Psi(t)},$$

  where $\psi(t)$ is the PDF and $\Psi(t)$ is the CDF. Instead of border infection probability, we compute the average infection rate on the border by taking the average of instantaneous rates at the turn of epochs.

- "homogeneous rates": Similar solution as above.

- "SIS epidemics": Consider a different compartmental model with $k$ compartments. Then instead of a number, we can send a PMF, summarizing the probability of a border vertex being in different compartments. Then instead of one integer, we send $k - 1$ integers.

- "a fixed graph": We can allow the graph to change at the end of each epoch, an idea also adopted in [10].

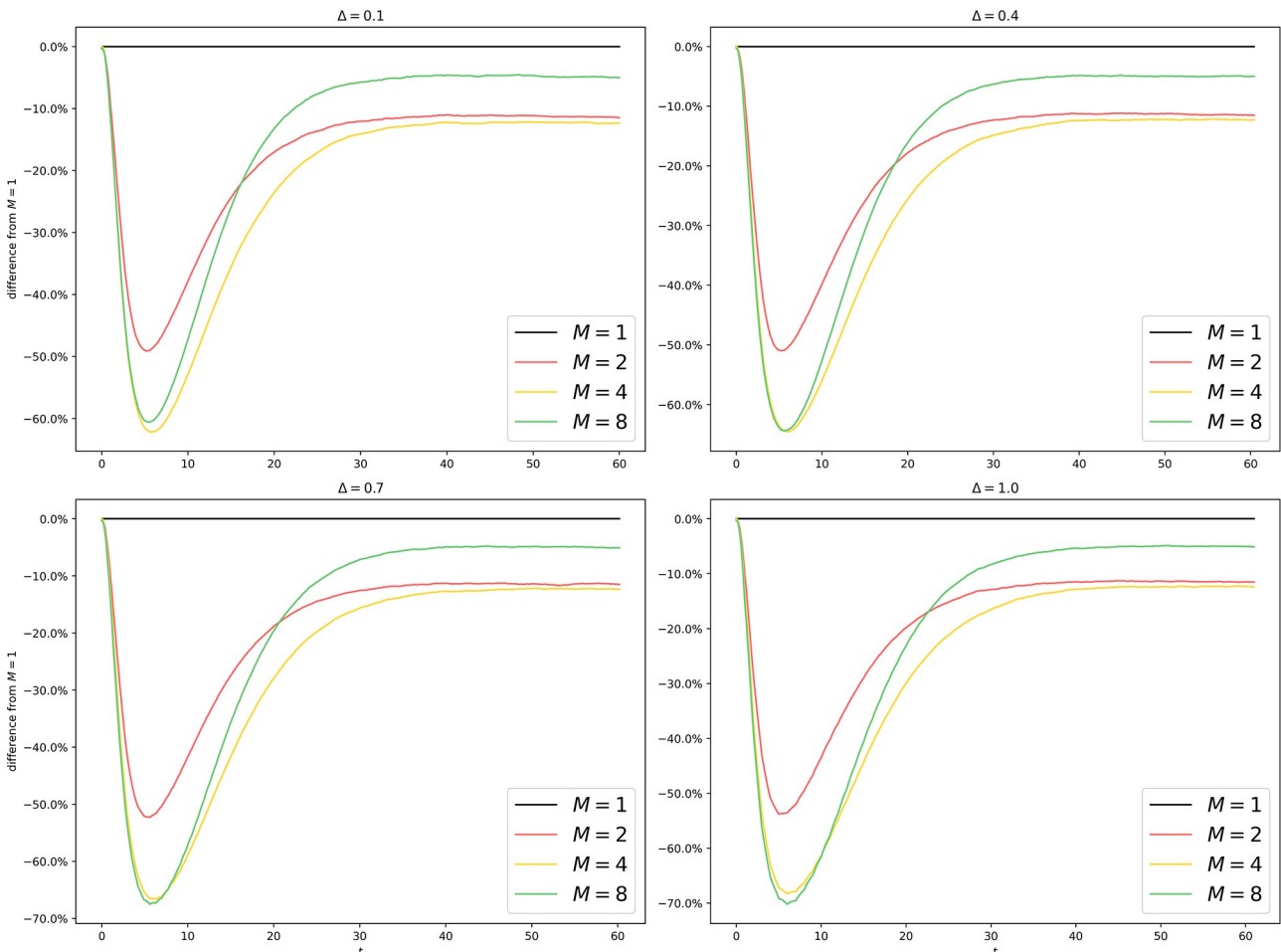

**Fig 12. Uniform partitioning of dblp260k: Difference from _M_ = 1 with different _M_ and Δ.** A uniform partitioning of dblp260k yields highly unsatisfactory results with a type-2 error of more than 10%, indicating that while a naive uniform partitioning scheme works well for naive Erdos-Renyi graphs, sophisticated graph structures demand sophisticated partitioning schemes.

Moreover, our approach of sending the border infection probability is just one of the many ways of sending a summary of this process to another. This approach has two main downsides. We discuss them below and offer some possible remedies.

- It allows infection events to occur where there should be none. Consider a vertex _v_ on _P_ and a vertex _u_ on _Q_, each being the only neighbor of the other. Suppose _v_ and _u_ are susceptible at the beginning of the epoch. As long as some infected vertices exist on the (_P_, _Q_) border or the (_Q_, _P_) border, _u_ or _v_ can turn infected by the end of the epoch, even though neither has any infected neighbors. A solution to this problem is to send the state of the whole border over the network, although this will inevitably increase bandwidth usage.

- Whether we send the state of the whole border or just a summary statistic, the information is stale. One expedient is to use a smaller Δ, limiting how stale the information can be. Another solution is to "predict" the state dynamics of the remote process, even though this is more of an art than a science and not easily generalizable to applications beyond epidemiology.

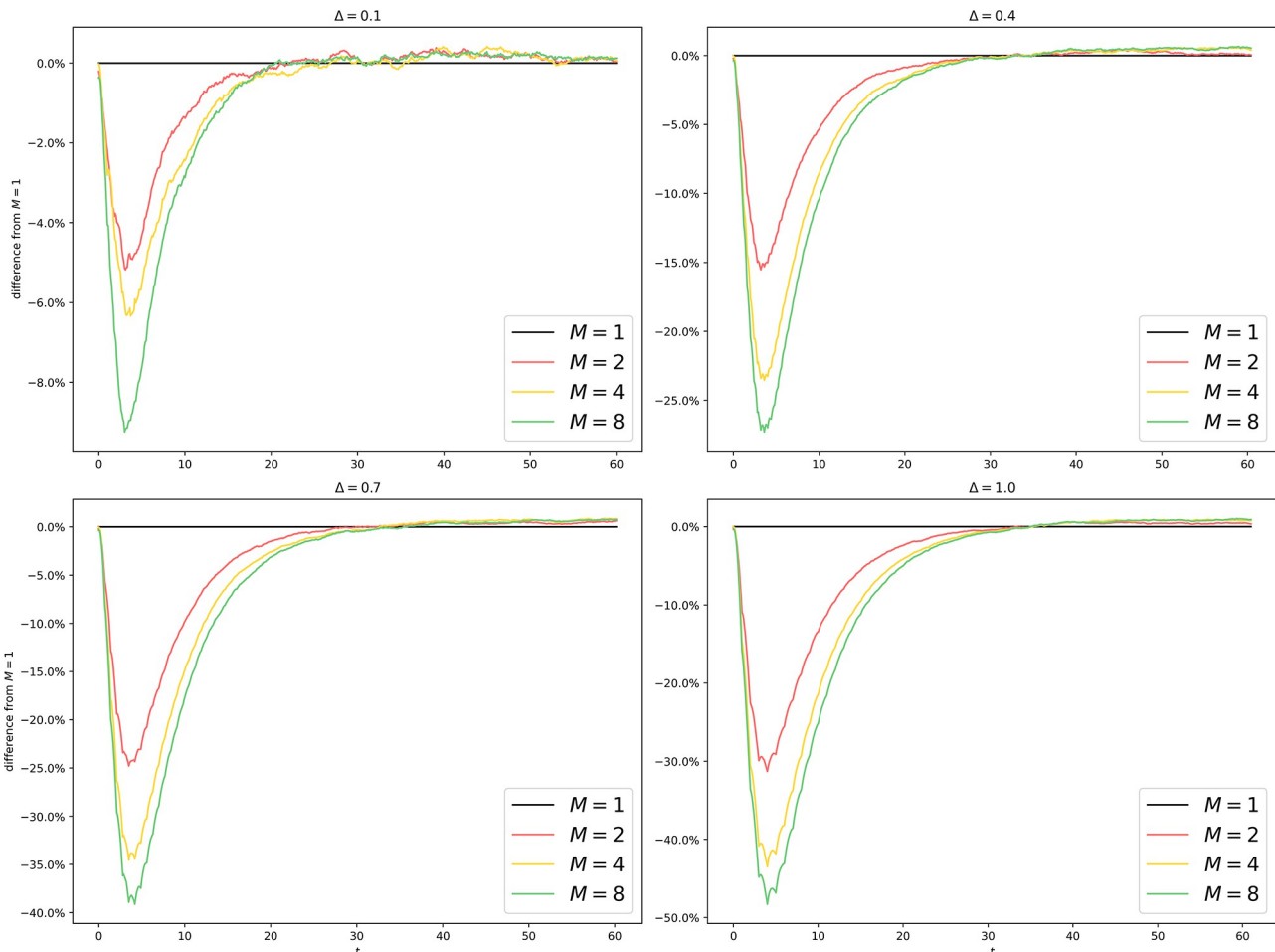

**Fig 13. Uniform partitioning of dblp260k (GCI): Difference from $M$ = 1 with different $M$ and $\Delta$.** The GCI brings significant improvements compared to Fig 12 by bringing the type-2 error down to less than 1%.

## Supporting information

**S1 File.**
(PDF)

## Author Contributions

**Conceptualization:** Guohao Dou.

**Software:** Guohao Dou.

**Validation:** Guohao Dou.

**Visualization:** Guohao Dou.

**Writing – original draft:** Guohao Dou.

**Writing – review & editing:** Guohao Dou.

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
