## [Decision Letter · Decision Letter 0]

24 Jul 2023

PONE-D-23-19886Scalable parallel and distributed simulation of an epidemic on a graphPLOS ONE

Dear Dr. Dou,

Thank you for submitting your manuscript to PLOS ONE. After careful consideration, we feel that it has merit but does not fully meet PLOS ONE’s publication criteria as it currently stands. Therefore, we invite you to submit a revised version of the manuscript that addresses the points raised during the review process.

We look forward to receiving your revised manuscript.

Kind regards,

Wei Ju, Ph.D.

Academic Editor

PLOS ONE

Journal Requirements:

Reviewers' comments:

Reviewer's Responses to Questions

**Comments to the Author**

1. Is the manuscript technically sound, and do the data support the conclusions?

Reviewer #1: Yes

Reviewer #2: Yes

2. Has the statistical analysis been performed appropriately and rigorously? 

Reviewer #1: Yes

Reviewer #2: Yes

3. Have the authors made all data underlying the findings in their manuscript fully available?

Reviewer #1: Yes

Reviewer #2: No

4. Is the manuscript presented in an intelligible fashion and written in standard English?

Reviewer #1: Yes

Reviewer #2: Yes

5. Review Comments to the Author

Reviewer #1: This paper proposes an algorithm to simulate Markovian SIS epidemics with homogeneous rates and pairwise interactions on a fixed undirected graph, assuming a distributed memory model of parallel programming and limited bandwidth. This paper is well organized and clearly written. The technical details are also easy to follow. The experiment are extensive and helpful to validate the effectiveness of the model. However, I have the following concerns:

1. There is a lack of clarity in Figures 1 and 2 pipeline-like scheme would provide a better understanding of the model's mechanisms.

2. The novelty of the proposed technique can be presented better for understanding the readers.

3. Authors should summarize their experiment results in the abstract.

4. Can the proposed method be combined with graph neural networks [1]. Please include some discussion.

[1] A Comprehensive Survey on Deep Graph Representation Learning, arxiv 2023

Reviewer #2: Summary:

The paper proposes an algorithm for Markovian SIS (Susceptible-Infected-Susceptible) epidemics modeling and simulations using graphs for distributed and parallel computing setups. The method combines the advantages of two existing techniques, Time-Dependent Simulation (TDS) and Event-Driven Simulation (EDS), to handle the challenging task of building large graph-based parallel models for global pandemic simulation. The proposed algorithm involves event-driven local computations and summary communications of boarding processes. The paper includes theoretical analysis and quantitative experiments to demonstrate the effectiveness of the method.

Strengths:

One of the major strengths of this paper lies in its direct applicability to real-world public health issues, which are gaining unprecedented attention in recent times. By tackling this pressing problem, the paper offers a valuable contribution to the field.

Moreover, the algorithm itself introduces a novel approach by integrating the benefits of both TDS and EDS, showcasing an innovative solution for pandemic simulations.

The inclusion of theoretical proofs and pseudocode enhances the reproducibility of the proposed method.

Weaknesses:

One concern is that the real dataset used for evaluation comes from a social network, not global epidemics, raising doubts about its suitability and generalizability for epidemic simulation.

Furthermore, the paper could benefit from providing more comprehensive background knowledge on susceptible-infected-susceptible models. Introducing key concepts and variables with proper explanations when they first appear would make the paper more accessible to a broader audience, including readers with varying levels of expertise. The paper provides direct comparisons with the TDS algorithm, but it does not include direct comparisons with the EDS method, which forms the basis of the proposed algorithm. The reasons behind this choice should be clarified by the author.

comments:

• Consider adding a "Related Work" section to provide context and comparisons with prior research.

• Consider introducing initial vertex states INF and SUS with a brief explanation.

• Consider renaming section 2 to more detailed titles such as “Problem Setup,” “Method,” “Proofs,” “Experiments,” “Discussion,” and “Results.”

• Consider adding a few words on the motivation in the abstract to engage readers.

• Suggest further polishing the paper and addressing any grammar errors to enhance the readability.

6. PLOS authors have the option to publish the peer review history of their article (what does this mean?). If published, this will include your full peer review and any attached files.

Reviewer #1: No

Reviewer #2: **Yes: **ANWAR SAID

---

## [Author Response · Author response to Decision Letter 0]

22 Aug 2023

Thanks for reviewing my article. Please kindly find the response letter among the attached files with name "Response to Reviewers". 

Have a great day!...

---

## [Decision Letter · Decision Letter 1]

7 Sep 2023

Scalable parallel and distributed simulation of an epidemic on a graph

PONE-D-23-19886R1

Dear Dr. Dou,

We’re pleased to inform you that your manuscript has been judged scientifically suitable for publication and will be formally accepted for publication once it meets all outstanding technical requirements.

Kind regards,

Wei Ju, Ph.D.

Academic Editor

PLOS ONE

Additional Editor Comments (optional):

Reviewers' comments:

Reviewer's Responses to Questions

**Comments to the Author**

1. If the authors have adequately addressed your comments raised in a previous round of review and you feel that this manuscript is now acceptable for publication, you may indicate that here to bypass the “Comments to the Author” section, enter your conflict of interest statement in the “Confidential to Editor” section, and submit your "Accept" recommendation.

Reviewer #1: All comments have been addressed

Reviewer #2: All comments have been addressed

2. Is the manuscript technically sound, and do the data support the conclusions?

Reviewer #1: Yes

Reviewer #2: Yes

3. Has the statistical analysis been performed appropriately and rigorously? 

Reviewer #1: Yes

Reviewer #2: Yes

4. Have the authors made all data underlying the findings in their manuscript fully available?

Reviewer #1: Yes

Reviewer #2: Yes

5. Is the manuscript presented in an intelligible fashion and written in standard English?

Reviewer #1: Yes

Reviewer #2: Yes

6. Review Comments to the Author

Reviewer #1: This paper is about scalable parallel and distributed simulation of an epidemic on a graph. Authors have addressed all my concerns. I lean to accept.

Reviewer #2: The authors have carefully addressed all my comments. The paper seems complete to me now and I have no further questions.

7. PLOS authors have the option to publish the peer review history of their article (what does this mean?). If published, this will include your full peer review and any attached files.

Reviewer #1: No

Reviewer #2: **Yes: **Anwar Said

---

## [Editor Report · Acceptance letter]

20 Sep 2023

PONE-D-23-19886R1 

Scalable parallel and distributed simulation of an epidemic on a graph 

Dear Dr. Dou:

I'm pleased to inform you that your manuscript has been deemed suitable for publication in PLOS ONE. Congratulations! Your manuscript is now with our production department. 

Kind regards, 

on behalf of

Dr. Wei Ju 

%CORR_ED_EDITOR_ROLE%

PLOS ONE